# Xanthine Dehydrogenase Is a Modulator of Dopaminergic Neurodegeneration in Response to Bacterial Metabolite Exposure in *C. elegans*

**DOI:** 10.3390/cells12081170

**Published:** 2023-04-15

**Authors:** Jennifer L. Thies, Karolina Willicott, Maici L. Craig, Madeline R. Greene, Cassandra N. DuGay, Guy A. Caldwell, Kim A. Caldwell

**Affiliations:** 1Department of Biological Sciences, The University of Alabama, Tuscaloosa, AL 35487, USA; 2Center for Neurodegeneration and Experimental Therapeutics, Department of Neurology, Heersink School of Medicine, University of Alabama at Birmingham, Birmingham, AL 35294, USA

**Keywords:** CYP2, UGT, C25D7.5, XO, *Streptomyces venezuelae*, bioactivation, calcium, *daf*-*16*, *pqm*-*1*, PI16, *scl*-*24*, *scl*-*25*

## Abstract

Oxidative stress is a contributing factor to Parkinson’s disease (PD). Considering the prevalence of sporadic PD, environmental exposures are postulated to increase reactive oxygen species and either incite or exacerbate neurodegeneration. We previously determined that exposure to the common soil bacterium, *Streptomyces venezuelae* (*S. ven*), enhanced oxidative stress and mitochondrial dysfunction in *Caenorhabditis elegans*, leading to dopaminergic (DA) neurodegeneration. Here, *S. ven* metabolite exposure in *C. elegans* was followed by RNA-Seq analysis. Half of the differentially identified genes (DEGs) were associated with the transcription factor DAF-16 (FOXO), which is a key node in regulating stress response. Our DEGs were enriched for Phase I (CYP) and Phase II (UGT) detoxification genes and non-CYP Phase I enzymes associated with oxidative metabolism, including the downregulated xanthine dehydrogenase gene, *xdh*-*1*. The XDH-1 enzyme exhibits reversible interconversion to xanthine oxidase (XO) in response to calcium. *S. ven* metabolite exposure enhanced XO activity in *C. elegans*. The chelation of calcium diminishes the conversion of XDH-1 to XO and results in neuroprotection from *S. ven* exposure, whereas CaCl_2_ supplementation enhanced neurodegeneration. These results suggest a defense mechanism that delimits the pool of XDH-1 available for interconversion to XO, and associated ROS production, in response to metabolite exposure.

## 1. Introduction

Parkinson’s Disease (PD) is the second most common neurodegenerative disorder. Despite the prevalence, genetic contributors and risk factors explain a fraction of cases [1]. Therefore, to advance our understanding of PD, an exploration of exposures is warranted. For example, herbicides, such as paraquat, and the pesticide rotenone, induce neuronal degeneration and contribute to parkinsonism in humans [2,3,4]. These toxicants have provided the foundation for understanding that oxidative damage leads to dopaminergic (DA) neurodegeneration. For example, rotenone directly inhibits mitochondrial complex I [5,6]. In addition to chemical exposures, environmental risks can lead to neurodegeneration. Likewise, increasing evidence worldwide implicates exposure to a derivative of the amino acid alanine, β-Methylamino-Alanine (BMAA), found in cycads and cyanobacteria, as a cause of amyotrophic lateral sclerosis syndrome associated with parkinsonism [7].

Advances in genomic analysis have profoundly increased our understanding of neurodegenerative disease mechanisms and expanded prospects for therapeutic intervention. Nevertheless, the specific gene-by-environment interactions widely attributed to the causation and/or increased risk for neurodegenerative diseases remain elusive. While definitive environmental contributors to PD largely remain undefined, one well-established and reproducible association to increased disease prevalence is living in a rural area. Epidemiological analyses revealed that drinking well water, farming, a rural residence, and exposure to pesticides or herbicides may all be risk factors for developing PD [8,9,10]. Individuals living in rural environments typically experience more interaction with the terrestrial environment, whether by necessity (dirt floors, drinking well water) or by choice (avocation, occupation). A single gram of soil has been shown to contain as many as 1 billion microorganisms [11]. Within the order *Actinomycetales*, the ubiquitous soil bacterial genus *Streptomyces* contributes ~6% to this total [12]. These Gram-positive, aerobic organisms are responsible for producing greater than 70% of known antibiotics [13] in addition to a suite of other metabolites. At least four characterized proteasome inhibitors are products of *Streptomycetes* isolated from soil, including lactacystin [14]. We, therefore, hypothesized that enhanced exposure to these common soil bacteria might contribute to the onset or progression of neurodegeneration [15].

We previously identified a secondary metabolite produced by the common soil bacterium, *Streptomyces venezuelae* (*S. ven*), that caused age- and dose-dependent neurodegeneration in *C. elegans* neurons and the death of SH-SY5Y neurons in cell culture [16]. *S. ven* bacteria were grown to a stationary phase in a liquid media. The excreted secondary byproduct was separated through centrifugation and the collection of the spent medium. It was then extracted with dichloromethane (DCM) and dried under nitrogen gas. The neutral-lipid metabolic fraction of interest was collected through rotation evaporation to remove any residual DCM and then resuspended in ethyl acetate (EtAc). Activity levels were confirmed using DA neuron death assays in *C. elegans*, where worms were exposed to EtAc at a final concentration of 0.6% and seeded on top of the bacteria on the media. Through genetic and cellular characterization, we previously determined that exposure to the metabolite increased ROS production and impaired mitochondrial function in *C. elegans* [15,17,18].

In this current follow-up study, we exposed worms to the *S. ven* metabolite and generated a transcriptional profile using RNA-sequencing (RNA-seq) to discover genes associated with the neurotoxic response to the metabolite. Through subsequent functional analysis by RNA interference (RNAi) in combination with bioassays for quantitative assessment of dopaminergic neurodegeneration, we discerned the impact of differentially expressed genes (DEGs) in worms treated with *S. ven* metabolite in a temporal manner.

Outcomes of this strategy included the identification of DEGs that are primarily associated with stress response and xenobiotic detoxification. Notably, nearly half the differentially expressed transcripts showed a correlation with upstream sequence signatures associated with DAF-16 (FOXO), a critical modulator of lifespan and cellular stress response in *C. elegans* [19]. Among the regulated transcripts were several Phase I (cytochrome P450) and Phase II [uridine 5′-diphospho-glucronosylatransferase (UGT)] gene products. Phase I reactions typically function to increase compound solubility and reduce toxicity. However, we determined that components of both the Phase I and the Phase II reactions enhanced the neurotoxicity observed in *C. elegans* from *S. ven* metabolite exposures in a manner consistent with bioactivation [20,21,22].

The transcriptional response of *C. elegans* to *S. ven* exposures also revealed significant changes in the expression of non-CYP oxidative enzymes, including alcohol dehydrogenase (*adh*-*1*) and xanthine dehydrogenase (*xdh*-*1*). Interestingly, the XDH protein has been characterized as performing distinct functions through interconversion to xanthine oxidase (XO). Physiologically, both enzymes can participate in the detoxification of xenobiotics and endogenous compounds; however, the transition to XO also results in an increase in ROS production [23]. In this context, we experimentally confirmed that XO enzyme activity was increased in animals treated with *S. ven* metabolite and that enzymatic interconversion exhibited a dependence on calcium levels. Likewise, the neurotoxic activity of the *S. ven* metabolite was significantly reduced in the absence of calcium supplementation. Given the functional prominence of Phase I and II metabolic gene regulation in organismal defenses from toxicants, as well as previously reported interactions between XO and calcium [24,25,26], the results of this study collectively support a mechanism whereby the *S. ven* metabolite triggers a calcium-dependent cellular stress response in *C. elegans* neurons. The increased oxidative stress and ROS, concomitant with changes in gene expression, render animals susceptible to dopaminergic neurodegeneration [17,18].

## 2. Materials and Methods

### 2.1. C. elegans Strains

Nematodes were grown and maintained on OP50 *E. coli* bacteria at 20 °C using standard *C. elegans* laboratory conditions [27]. Some strains were provided by the Caenorhabditis Genetics Center (CGC), which is funded by the NIH Office of Research Infrastructure Programs (P40 OD010440). The following strains were obtained from the (CGC): N2 (Bristol), CF1038 (*daf*-*16(mu86*)) and RB711 (*pqm*-*1*(*ok485*)). We obtained BY250 (*vtls7*[P*_dat-1_*::GFP]) from Randy Blakely (Florida Atlantic University, Boca Raton, FL, USA).

### 2.2. Isolation and Extraction of S. venezuelae Metabolite

*Streptomyces venezuelae* (*S. ven*) metabolite was generated as previously described [17]. Briefly, spores from *S. ven* (ARS NRRL) were inoculated in 6 L of SYZ media in artificial saltwater and grown at 30 °C in a floor shaker for 3 weeks. Samples were then collected, and the cell debris was removed by centrifugation at 12,000× *g* for 10 min; supernatants were sequentially passed through eight PES filter membranes with the following range of pore sizes: 6, 2.7, 2.0, 1.6, 1.2, 0.7, 0.45 and 0.22 μm. The retrieved conditioned media was further extracted with an equal volume of dimethylchloride (DCM) 4 times using a separatory funnel. The DCM organic layers were collected and dried, and the remaining residue was resuspended in 2 mL of EtAc to create a concentrated stock solution. New batches of metabolite were tested for neurodegenerative activity, where 25 μL of concentrated stock solution was reconstituted in 2 mL of EtAc to establish a working concentration. This “20×” concentration was assayed for neurodegeneration, as described in the next section. If this concentration resulted in ~20% death of DA neurons by day 9 of exposure, then this was used as the working stock for all assays. If not, then further titration was performed. For the experiments in this manuscript, one batch of *S. ven* metabolite was used for all experiments except for the xanthine oxidase and the *daf*-*16* RT-qPCR experiments. We operationally define *S. ven* activity as a neurotoxic “metabolite”. However, the extract we use is partially purified, based on HPLC chromatography, and represents a mixture of several molecules.

### 2.3. S. venezuelae Metabolite Treatment

For worm exposures, 60 μL of the 20× concentration of metabolite was added to the surface of the bacterial lawn on nematode growth medium (NGM) Petri plates (60 mm diameter) and allowed to dry in a biological safety hood for 15 min. The 20× *S. ven* consists of 25 μL of concentrated stock solution reconstituted in 2 mL of EtAc. Control plates received 60 μL of EtAc solvent (final concentration of 0.6%). Animals were transferred to freshly applied bacterial plates every day and were continually exposed to the metabolite or solvent from hatching until the day of RNA extraction.

### 2.4. C. elegans RNA Isolation Paradigm and Sample Preparation

To determine the appropriate days for RNA extraction following *S. ven* metabolite treatment to *C. elegans*, we examined former outcomes of exposure. Firstly, the *S. ven* metabolite significantly upregulated *hsp*-*6*::GFP expression in day 4 worms [17]. HSP-6 is a nuclear-encoded mitochondrial chaperone that reacts to ROS via the mitochondrial unfolded protein response pathway (UPR^mt^). This pathway is cytoprotective when it is activated in response to acute stressors where it promotes cell survival and organelle recovery [28], but it can also be damaging if dysregulated, leading to cell death [29]. Secondly, by examining mitochondrial oxidative stress using the *sod*-*3*::GFP reporter strain [30], increased GFP fluorescence was observed by day 5 of *S. ven* exposure [17]. Finally, the metabolite reproducibly induced progressive dopaminergic neurodegeneration, where ~15% and 40% of the animals displayed degeneration at days 7 and 9, respectively [16,18]. These readouts led to our selection of days 5 and 7 for RNA isolation.

To prepare worms for extraction, animals were grown on standard nematode growth medium (NGM) seeded with *E. coli* strain OP50 for two generations to ensure that crowding and starvation did not impact gene expression data. Briefly, twelve 60 mm NGM plates were used for each treatment. The plates were spotted with 200 μL of *E. coli*, dried in a biological safety hood and allowed to grow at 37 °C O/N. The plates were then overlayed with 60 μL of 20× *S.ven* metabolite and allowed to dry in a chemical safety cabinet for 20 min before 10–15 adult hermaphrodites were allowed to egg lay for 3 h. Adult hermaphrodites were removed, and progeny were allowed to grow up at 20 °C until the day of the extraction. On the first day of isolation, 35 worms were collected from each plate and transferred onto an unseeded plate for a total of 420 worms. The remaining 115 worms were transferred to newly layered *S. ven* plates to allow worms to grow up for the later isolation days. The same was also undertaken for the ethyl acetate control population. Additionally, this collection method was used for RNA extraction and isolation on days 5 and 7. To ensure that the *S. ven* metabolite was still active and continued causing neurodegeneration in the animals that we extracted RNA from, a small subset of 10 worms from each plate were collected following day 7, and transferred until day 9, at which time dopaminergic neurodegeneration analysis was completed. Replicates in which no dopaminergic neurodegeneration was seen were removed from the experiment, and additional replicates were performed.

### 2.5. RNA Isolation, Extraction

All materials used for the isolation and extraction of RNA were treated with DEPC and autoclaved to ensure that they did not contain contamination from any external sources. The total RNA was extracted from *C. elegans* populations using the RNeasy Micro Kit (Qiagen, Germantown, MD, USA). ~400 nematodes were moved onto an unseeded NGM plate and washed into an RNase free 10 mL glass conical tube with 6 mL of RNase free 0.5× M9. The worms were washed 4 times with RNAse-free 0.5× M9, which comprises of centrifuging the glass tube at 1.6 rpm for 2 min, removing the supernatant without disturbing the pellet, adding fresh 0.5× M9 into the tube and gently inverting the tube to resuspend the pellet. Following the fourth wash, the worms were resuspended in 5 mL of RNAse-free 0.5× M9 and allowed to rock on a shaker at the 3.5 setting for 20 min to purge and clear the worm digestive track of any bacterial sources. Following the purge, glass conical tubes were spun down at 1.6 rpm for 2 min and subsequently washed two more times with RNAse-free double-distilled H_2_O. Following the final wash, the supernatant was carefully removed without disturbing the pellet, so the remaining volume in the tube was approximately 300 μL. The remaining mixture of worms was transferred to a low-bind RNAse-free microcentrifuge tube and centrifuged at 1.6 rpm for 2 min. After this centrifugation cycle, as much supernatant as possible was removed without disturbing the worm pellet. To the pellet, 500 μL of Trizol reagent was added and vortexed for 30 s. This mixture then went through a series of 7 freeze–thaw cycles in liquid nitrogen and then 8 freeze–vortex cycles before standing at RT for 5 min. Following the 5 min wait, 100 μL of chloroform was added to each sample and inverted continuously for 15 s. After inversion, samples were placed at RT to allow for phase separation.

### 2.6. RNA Quality Control (QC)

*C. elegans* RNA samples were immediately frozen at −80 °C following extraction. Small aliquots of each sample were frozen separately for QC analysis at the completion of each replicate. 3 μL of RNA was saved for spectrophotometer analysis for RNA concentration (ng/mL) and purity was performed on a Nanodrop (Thermo Fisher Scientific, Waltham, MA, USA). A 260/280 ratio between 1.9 and 2.1 and a 260/230 ratio above 1.9 were our preferred parameters. This sample was also used to confirm RNA quality using the Agilent Bioanalyzer and Qubit systems. A 4 μL sample was saved to run on a 1% agarose gel to identify samples that had protein contamination as well as RNA integrity.

Bioinformatic analysis RNA samples were sent to Novogene Co. in Sacramento, CA, USA. Additional quality control (QC) was performed by Novogene to ensure that the samples were pure, uncontaminated, and undegraded. All samples passed QC, and cDNA libraries were created from our RNA samples. Two paired-end 150 bp reads were sequenced on an Illumina HiSeq. The data were filtered by removing low-quality and adaptor-contaminated reads. Bioinformatic analysis was performed in-house using Salmon v0.11.3 [31]. First, a transcriptome join file was created by concatenating the N2 reference cDNA and ncRNA datasets obtained from Ensembl (http://ftp.ensembl.org/pub/release-99/fasta/caenorhabditis_elegans/cdna/Caenorhabditis_elegans.WBcel235.cdna.all.fa.gz (accessed on 1 April 2020) and http://ftp.ensembl.org/pub/release-99/fasta/caenorhabditis_elegans/ncrna/Caenorhabditis_elegans.WBcel235.ncrna.fa.gz) (accessed on 1 April 2020). Next, the join file was prepared for quantifying with the “salmon index” command in the mapping-based mode. The transcripts were then quantified using the “salmon quant” command. The validateMappings flag enabled selective alignment during the mapping using an algorithm that detects redundancies. Finally, the quantified transcripts were annotated with the N2 reference GTF file (http://ftp.ensembl.org/pub/release-99/gtf/caenorhabditis_elegans/Caenorhabditis_elegans.WBcel235.99.gtf) (accessed on 1 April 2020) using the geneMap flag. The resultant file contained gene-level quantification, which was needed for the analysis of DEGs. DEGs were extracted using DESeq2 version 1.24.0 [32] in R. Significance was determined if the *p*-adjusted value was less than 0.057. DESeq2 utilized a Benjamini–Hochberg method of normalization to calculate the adjusted *p*-value, a powerful method for controlling the false discovery rate. Heatmaps were created in R using package “pheatmap” [33].

### 2.7. Differentially Expressed Gene (DEG) Analysis, Selection, and Functional Annotation

DEGs were selected using an adjusted *p*-value of 0.057. These genes were then categorized by fold change. Genes that had both significant adjusted *p*-values and a positive log2 fold change were used for further analysis. Gene ontology (GO) analyses were created using the WormCat bioinformatics resource [34]. Genes that were significantly up- or down-regulated, as determined by an adjusted *p*-value of less than *p* < 0.057, were used in the analysis.

### 2.8. DAF-16 Binding Motif Analysis

The 1 kb upstream sequences of each DEG were obtained from Wormbase and then analyzed using the Regulatory Sequence Analysis Tool (RSAT) (http://rsat.sb-roscoff.fr) accessed on 1 October 2022. This tool searched for the canonical DAF-16-binding element [DBE (TTGTTTAC)] and DAF-16-associated element [DAE (CTTATCA)] sequences within our target genes. The *p*-value for comparing DAF-16 binding motif frequency between our DEGs and the whole *C. elegans* genome was calculated using the GIGA calculator (https://www.gigacalculator.com/calculators/p-value-significance-calculator.php) accessed on 1 October 2022 with a *p*-value set to <0.05.

### 2.9. Neurodegeneration Analysis and Fluorescence Microscopy

Animals for analysis were synchronized with a 3 h egg-lay using day 4 gravid hermaphrodites and incubated at 20 °C. *C. elegans* were raised on standard nematode growth media (NGM) plates, which have 1 mM CaCl_2_ added to the media. This was standard for all experiments in the manuscript except for when noted in Figure 6, when different concentrations of calcium were examined for an impact on DA neurodegeneration (i.e., 2 mM, 3 mM, and no calcium added to the medium). For all experiments in the manuscript, on the day of analysis, 30–35 worms were picked and immobilized in 10 mM levamisole on glass coverslips, inverted, and mounted on 2% agarose pads on microscope slides. Three total replicates were completed for each analysis, and 30 animals were scored per treatment (30 animals × 3 replicates = 90). Worms were scored for dopaminergic neurodegeneration on day 9 post-hatching. An animal was scored as normal (or WT) if it contained all 6 anterior DA neurons. If a worm displayed any degenerative phenotype, such as a missing process, severe blebbing of the process or cell body loss, it was scored as degenerative. Fluorescent microscopy was used for the neurodegeneration assay using a Nikon Eclipse epifluorescence microscope equipped with an Endow GFP HYQ filter cube (Chroma Technology, Bellows Falls, VT, USA). A cool Snap CCD camera (Photometrics, Tuscan, AZ, USA) driven by MetaMorph software. Metamorph NX 2.0, Meta Series Software v.7.8.9 (Molecular Devices, Sunnyvale, CA, USA) was used to acquire the images.

### 2.10. RNA Interference (RNAi) Experiments

RNAi feeding constructs were obtained from the *C. elegans* Ahringer library [35] or from Dharmacon as open reading frame constructs (ORFs) that were transformed into the L4440 destination vector and subsequently transformed into HT115 cells. To ensure these constructs contained the right mutation, clones were sequenced to verify accuracy. RNAi feeding clones were cultivated initially on LB solid media containing tetracycline (15 μg/mL) and ampicillin (100 μg/mL), and then individual colonies were grown overnight in liquid LB media containing ampicillin (100 μg/mL). IPTG was added to NGM plates for a final concentration of 1 mM, plated with RNAi feeding clones, and allowed to dry. Plates were induced overnight at 20 °C. Fertile adult hermaphrodites were allowed to lay eggs for 3–4 h to obtain a synchronized population. Plates were seeded with either 60 μL EtAc or 60 μL 20× *S. ven* metabolite until the day of analysis for all experiments, as described in the figure legends.

### 2.11. RNA Extraction and Reverse Transcription Real Time Quantitative PCR

These reactions were performed using IQ SYBR Green Supermix (Bio-Rad, Hercules, CA, USA) with the CFX96 Real-Time System (Bio-Rad), as described previously [36]. Worms were grown on standard NGM plates +60 μL EtAc or 60 μL 20× *S. ven* metabolite treatments. For the validation RT-qPCR studies, genes were analyzed on the day they were identified in the transcriptomic work on either day 5 (*scl*-*24*, *scl*-*25*, *cyp*-*35 B2*) or day 7 (*scl*-*24*, *cyp*-*35B2*, *ugt*-*22*, *xdh*-*1*). For the *daf*-*16* and *pqm*-*1* mutant RT-qPCR experiment, RNA was extracted from both day 5 (*cyp*-*34A10* and *fmo*-*3*) and day 7 (*cyp*-*35B1* and *set*-*18*) adult hermaphrodites, depending on when the gene was identified in the RNAseq work. Worms were washed 3 times in RNase-free 0.5× M9, followed by a single wash in RNase-free water. The total RNA was isolated from 100 to 200 adult worms (BY250) from each independent sample using TRI-reagent (Molecular Research Center, Inc., Cincinnati, OH, USA) on day 5 or 7 from the associated RNAi construct or EV RNAi exposure or mutant animal exposure. Following DNase treatment (Promega, Madison, WI, USA), 1 μg of RNA was used to make complementary DNA (cDNA), which was synthesized with iScript Reverse Transcription Supermix for RT-qPCR (Bio-Rad, Hercules, CA, USA). PCR efficiency was calculated from standard curves that were generated using serial dilutions of the cDNA of all samples. All targeted genes were measured in triplicate. Amplification was not detected in non-template and non-reverse transcriptase controls. Each reaction contained: 7.5 μL of the iQ SYBR Green Supermix, (BioRad) 200 nM of forward and reverse primers, and 0.3 μL cDNA, to a final volume of 15 μL. Expression levels were normalized to three reference genes (*cdc*-*42*, *ama*-*1* and *pmp*-*3*) and were calculated using qBase^PLUS^ version 2.6 (Biogazelle, Gent, Belgium) to determine reference target stability. Three technical replicates were used for each sample. A one-tailed Student’s *t*-test was used for statistical comparison, and the data were presented as mean +/− standard error of the mean (SEM). Each primer pair was confirmed for at least 90–110% efficiency in a standard curve using BY250 cDNA. The primer pairs used were as follows. *cyp35B1*; Forward: CAAAGATGGAGCAGGAGAGG and Reverse: ATTGAATCCTGCGACCAAAG. *set-18*; Forward: GAGACACCGTTCGCCACT and Reverse: TGCCTGACACTCTTTACTACAATAC *cyp34A10*; Forward: ACAGCGGTGCACCTTCTACT and Reverse: CACCACATTTGGATGGTTCA. *fmo*-*3*; Forward: AGTGAAATGCAGGCGAGAGT and Reverse: ACCGAGTTCATGGAGGTACG. *scl*-*25*; Forward: TTGATTCGAAGGGGATCAAC and Reverse: GTTCCACACGAAGTCCCACT. *scl*-*24*; Forward: ACACACAATGCGCTGAAATC and Reverse: GAGCATGGCTCTCCTTTGTC. *cyp*-*32B2*; Forward: GAGAATCCGCATGATTTCGT and Reverse: GTGAGCCATTTTCCGTGATT. *ugt*-*22*; Forward: GCCGTTATGGTTCCCCTATT and Reverse: CGCATTTCCCAAATCAGTTT. *xdh*-*1*; Forward: TCATGAGATGCTCCATTGGT and Reverse: GGAGATGCTGTTGCAATGTT. *cdc*-*42*; Forward: CTGCTGGACAGGAAGATTACG and Reverse: CTCGGACATTCTCGAATGAAG. *pmp*-*3*; Forward: GTTCCCGTGTTCATCACTCAT and Reverse: ACACCGTCGAGAAGCTGTAG. *ama*-*1*; Forward: TCCTACGATGTATCGAGGCAA and Reverse: CTCCCTCCGGTGTAATAATGA.

### 2.12. Xanthine Oxidase Activity Measurement

Xanthine Oxidase Activity was measured using the Xanthine Oxidase Activity Kit (Sigma-Aldrich Inc, St. Louis, MO, USA). Worms were raised on standard NGM plates and exposed to 60 μL EtAc or 60 μL *S. ven* metabolite treatments for the duration of the experiment until Day 7 of adulthood. Similarly, for the no calcium experiment, worms were grown on NGM plates without the addition of CaCl_2_ until Day 7 of adulthood. At this time, for both experiments, ~150–200 adult hermaphrodites were collected in RNase-free lo-bind tubes and washed three times in M9. During each experiment, every sample went through three freeze/thaw cycles. Samples were subsequently sonicated using an Ultrasonic Processor GEX 130 PB (VWR, Radnor, PA, USA) for three cycles of 10 s on and 30 s off at 40% amplitude. Samples were spun down at 15,000× *g* for 10 min. Furthermore, 50 μL of the sample was then placed into respective wells of a clear 96-well plate for analysis. The experiment was performed as per the directions in the kit for colorimetric analysis, and the assay was run at 570 nm for 60 min with readings taken every 5 min using a SpectraMax M2 e Microplate Reader (Molecular Devices, Sunnyvale, CA, USA).

### 2.13. EGTA Treatments

For both the xanthine oxidase activity measurement and the neurodegeneration assay, worms were grown in the presence of 0.5 mM EGTA for the first 3 days (embryo to 1-day adulthood). EGTA was seeded on top of the OP50 bacteria and allowed to dry for 10–15 min, then seeded with 60 μL EtAc or 60 μL *S. ven* metabolite treatments. After this initial exposure, worms were transferred to standard NGM plates until the appropriate day of analysis corresponding to the specific assay.

### 2.14. Statistical Analysis

Statistical analysis was performed using Prism 9 software (GraphPad Software Inc., La Jolla, CA, USA). The differences between the treatments, groups, and conditions were measured by Student’s *t*-test, one-way ANOVA, or two-way ANOVA. For qPCR validation experiments, a one-tailed Student’s *t*-test was used. If two days of validation were used, a two-way ANOVA with Sidak’s correction was used. For two-way ANOVA analyses, a full model fit for interaction was applied. Where appropriate, comparisons between EtAc and metabolite groups and candidate genes used Sidak’s or Tukey’s post hoc test, in addition to experiments that contained more than two conditions, for differences when *p* < 0.05. All neurodegeneration data are presented as mean +/− standard deviation (SD).

## 3. Results

### 3.1. Induction of Differential Gene Expression in C. elegans in Response to S. ven Metabolite 

To gain a better understanding of how the *S. ven* metabolite impacts gene expression changes in *C. elegans*, we performed RNA-seq to transcriptionally profile expression in exposed worms. The *S. ven* metabolite was produced as previously described [18]. Following confirmation of neurotoxic activity, the concentration of *S. ven* used in all assays was titrated so that it resulted in at least 20% death of DA neurons by day 9 of exposure. While we operationally define this activity as a neurotoxic “metabolite” from *S. ven* in this manuscript, the extract we use in our studies is partially purified, based on HPLC chromatographic separation information, and represents a mixture of several molecules. In our experience, EtAc exposure at the concentration used does not cause significant neurodegeneration or cellular stress in *C. elegans* [16,17,18,28]. To determine the appropriate days for RNA extraction following *S. ven* metabolite treatment to *C. elegans*, we retrospectively examined former phenotypic outcomes of exposure data, including the response of *S. ven*-treated animals to ROS and dopaminergic neurodegeneration [16,17,18]. These cellular readouts following *S. ven* metabolite treatment provided in vivo criteria for the systematic selection of days 5 and 7 as logical timepoints for RNA isolation. We predicted that inciting events leading to widespread oxidative stress and neurodegeneration in these phenotypically naïve animals would be logically identified within this temporal window.

Worms were chronically exposed to either *S. ven* metabolite or ethyl acetate (EtAc; vehicle control) from the embryo until the day of collection. To ensure that the metabolite was inducing neurodegeneration, we used transgenic worms that express GFP under the control of the dopamine transporter promoter (P*_dat_*_-*1*_::GFP; strain BY250) to highlight the dopamine neurons. The transcriptome from five biological replicates at two different time points (days 5 and 7) was characterized under each treatment condition (Figure 1). Within-group variation among the biological replicates for each treatment was less than between-group variation across the treatments. There was a strong separation between the treatments, and the variance between the replicates was small (Figure 2). Multivariate principal component analysis (PCA) indicated that highly reproducible gene expression changes occurred in worms exposed to the metabolite on both days of analysis. Specifically, PC1, which is associated with the variance between treatments, accounted for 72% of the explained variance on day 5, indicating that the treatments represent distinct differences from each other, while PC2 accounted for 12% on day 5, denoting low variation between the replicates (Figure 2A). On day 7, PC1 accounted for 80% of the explained variance, and PC2 accounted for 9% (Figure 2B), demonstrating even tighter clustering of the replicates.

Heatmap analysis revealed distinct gene regulation patterns in *C. elegans* in response to the *S. ven* metabolite compared to solvent control for both days 5 and 7 (Figure 2C,D). On day 5, 10 DEGs were identified (Table 1), while on day 7, 28 DEGs were identified, including three that were also upregulated on day 5 (Table 2). Notably, all but one of the day 5 DEGs (9/10) had corresponding human homologs [according to Wormbase (wormbase.org), v. WS286], while on day 7, identifiable human homologs were observed for 60% of the genes (17/28); 25 genes were significantly upregulated, and 10 were significantly downregulated (*p*adj < 0.0570) [Gene Expression Omnibus (GEO) Dataset Series: GSE217300]. In total, we identified 35 unique DEGs. Functional annotation and gene ontology enrichment of the DEGs were classified using WormCat [34]; stress response, metabolism and proteolysis were the most enriched categories (Table 1 and Table 2). When the DEGs were combined, significant enrichment was observed for proteolysis (five genes) and stress response (16 genes) (Table 3); these data suggest that the metabolite is being processed or modified by the worm either prior to or in response to the induction of organismal defenses.

### 3.2. Genes with DAF-16 Regulatory Motifs Are Enriched in an Age-Dependent Manner by the Metabolite

In *C. elegans*, the *daf*-*16* gene product encodes a well-studied transcription factor, DAF-16, which is an ortholog of human FOXO4 (forkhead box O4). DAF-16 regulates genes associated with redox homeostasis, detoxification, and stress response [37,38,39]. When cells are not stressed, DAF-16 is typically held inactive in the cytoplasm. However, upon encountering stress, DAF-16 becomes translocated to the nucleus, where it initiates the transcription of genes associated with diverse mechanisms, including longevity, stress, metabolism, and differentiation [40]. Notably, we previously observed that exposure to the *S. ven* metabolite increased the localization of DAF-16::GFP from the cytoplasm to the nucleus [17].

We examined the literature to determine which DEGs were experimentally validated as regulated by DAF-16 [37,41,42,43] and learned that 37% (14/38) of our DEGs had previous experimental evidence of an association with DAF-16 (Table 4). We then performed a sequence analysis to uncover potential DAF-16 regulatory motifs in the 1.0 Kb region upstream of the ATG start site of the DEGs. From this analysis, we uncovered eight more genes with DAF-16 regulatory motifs (Table 4). In total, 60% (23/38) of the DEGs had either a potential DAF-16 regulatory motif through our analysis or experimental evidence of DAF-16 regulation [42,43]; this enrichment was significant compared to the entire genome (*p* < 0.001).

The promoter sequences we identified for DAF-16 transcriptional regulation consisted of either the DAF-16-binding element (DBE) or the DAF-16-associated element (DAE) [34,44,45]. We detected four DEGs with DBE sequences (5′-TTGTTTAC-3′), which suggests these targets could be directly regulated by DAF-16 (Table 4), but this would need to be validated experimentally. There were also 12 DEGs with DAE sequences (5′-CTTATCA-3′), indicative of indirect regulation by DAF-16 [46]. It was previously shown that 21% of the ~20,000 *C. elegans* genes possess DAF-16-binding motifs in the 1.0 Kb upstream region [47]. However, no consensus in the literature exists as to how far upstream a binding site must be located. While most binding elements are found within the first 500 bp upstream from the start of the coding sequence, they have been identified as far away as 5 Kb [45]. Therefore, functional validation of DAF-16 in transcriptional regulation is necessary for more conclusive results.

### 3.3. DAF-16 and PQM-1 Affect Gene Expression in Response to S. ven Metabolite

We wanted to examine the impact of DAF-16 on the transcriptional regulation of select DEGs enriched in response to *S. ven* metabolite exposure containing predicted DBE or DAE motifs. We first examined whether DAF-16 exhibited a regulatory role of four select DEGs in a *daf*-*16* mutant background following *S. ven* metabolite exposure using RT-qPCR (Figure 3B–E). We first assessed the day 7 DEG, *cyp*-*35B1* (Table 2). It is a verified direct target of DAF-16 [41,42]. As expected for an upregulated DEG, when wild-type control worms (N2) were exposed to *S. ven*, *cyp-35B1* expression was significantly upregulated (Figure 3B). In *daf*-*16* mutants treated with *S. ven* metabolite, *cyp*-*35B1* displayed a significant reduction in expression compared to wild-type metabolite-treated animals (*p* < 0.01647), suggesting that DAF-16 might have a role in *cyp-35B1* expression (Figure 3B). We next analyzed the day 5 DEG, *cyp*-*34A10*, which was upregulated in our transcriptomics (Table 1). mRNA expression was measured in day 7 BY250 and *daf*-*16* mutant adult worms. In wild-type BY250 animals, exposure to *S. ven* resulted in significant upregulation of *cyp*-*34A10* expression (Figure 3C; *p* = 0.0104). In metabolite-treated *daf*-*16* mutants, *cyp*-*34A10* expression was substantially upregulated compared to wild-type levels (*p* < 0.0001). This implies that the presence of DAF-16 in wild-type animals functions to partially inhibit *cyp*-*34A10* expression (Figure 3C). Third, the *fmo*-*3* gene was an upregulated DEG at day 5 in our analysis (Table 1). In accordance with the RNA-seq data, *fmo*-*3* expression was significantly upregulated in N2 animals exposed to *S. ven* (Figure 3D). However, in *daf*-*16* mutants, *fmo*-*3* expression did not demonstrate a significant change from wild-type levels (*p* = 0.1251), indicating no contribution from DAF-16 toward *fmo*-*3* expression in response to metabolite (Figure 3D). Finally, we examined *set*-*18*; this day 7 DEG was a downregulated transcript in our RNA-seq analysis (Table 2). As expected, when worms were exposed to *S. ven*, *set*-*18* expression was significantly downregulated (Figure 3E; *p* = 0.0046). Strikingly, *set*-*18* exhibited a reversal of expression levels in the *daf*-*16* mutant background, as it became significantly upregulated compared to levels in wild-type treated animals (*p* = 0.0005) (Figure 3E).

We also monitored the transcriptional activity of the same DEGs in a *pqm*-*1* mutant background following *S. ven* metabolite exposure (Figure 3F–I). The PQM-1 transcription factor regulates the expression of genes downregulated in *daf*-*16* mutants and strongly correlates with DAE affinity [42,48]. Occasionally, DAF-16 and PQM-1 co-regulate a small subset of DAF-16 targets that contain both binding elements together [49]. *cyp*-*35B1* expression was not significantly different from control expression levels in *pqm*-*1* mutants treated with *S. ven* metabolite. This suggests that *pqm*-*1* does not modulate *cyp*-*35B1* expression (Figure 3F). Considering that *daf*-*16* depletion does not completely abolish the upregulation of *cyp*-*35B1* expression in response to a metabolite, an unidentified transcriptional modifier(s) likely exists. In contrast, when *pqm*-*1* mutants were treated with a metabolite, *cyp*-*34A10* expression was upregulated compared to wild-type levels (*p* = 0.0531), suggesting that PQM-1 acts to delimit *cyp*-*34A10* expression (Figure 3G). These data complement our sequence analysis, where were identified a DAE element in *cyp*-*34A10* (Table 4). Moreover, a previous report indicated positive binding affinities for both DAE and DBE using position-specific affinity scoring matrices [42]. In comparison to *daf*-*16* mutants, *pqm*-*1* mutants exhibited a decrease in *fmo*-*3* expression compared to wild-type levels (*p* = 0.0022), suggesting PQM-1 contributes to *fmo*-*3* expression following metabolite exposure (Figure 3H). Lastly, in *pqm*-*1* mutants, *set*-*18* expression in metabolite-treated worms was also significantly upregulated (*p* = 0.0004). This implies that *pqm*-*1* also has a role in *set*-*18* expression following metabolite exposure (Figure 3I). Notably, *set*-*18* was previously verified as a target of DAF-16 regulation [42,43]. This is consistent with our data; however, our data indicate that both DAF-16 and PQM-1 modulate *set-18* gene expression in response to *S. ven* metabolite exposure in *C. elegans*.

### 3.4. Knockdown of Innate Immunity-Associated DEGs Attenuates Metabolite-Induced DA Neurodegeneration 

Microbial-generated toxins can activate cellular surveillance pathways that are foundational for host survival [50,51,52]. Transcriptomic results revealed that several DEGs associated with innate immunity were upregulated in response to *S. ven* metabolite on days 5 and/or 7 (Table 1 and Table 2). To evaluate the influence of these gene products in whole animals, we used RNA interference (RNAi) to knock down these targets and assessed their impact on DA neurodegeneration following *S. ven* exposure in a cell non-autonomous manner (Figure 4A).

*C25D7.5* has not been characterized in *C. elegans*, but it is predicted to be localized to the plasma membrane and enable hydrolase and metal binding activity. The *C25D7.5* gene product has 44% amino acid sequence homology to human MBLAC1 (metallo-β-lactamase domain-containing protein 1). Importantly, a paralog of *C25D7.5* in *C. elegans*, *swip*-*10*, also contains a metallo-β-lactamase domain and induces dopaminergic neurodegeneration through a glutamate-specific mechanism [53]. The knockdown of *C25D7.5* rescues neurodegeneration when worms are exposed to *S. ven* compared to empty vector (EV) RNAi controls (*p* = 0.0161), indicating that the presence of *C25D7.5* contributes to metabolite-induced neurodegeneration (Figure 4B). A second upregulated DEG classified as functioning in innate immunity was *C. elegans vit*-*3*. This gene product is associated with yolk proteins that aid in the transport of lipids, and it is expressed in the nematode intestine. The human homolog is an Fc fragment of IgG-binding protein (FCGBP); it is thought to have a role in the innate immune system of the oral cavity and esophagus [54]. When knocked down, *vit*-*3* RNAi worms displayed less *S. ven*-induced neurodegeneration than the EV control animals (*p* = 0.0017) (Figure 4C).

Two DEGs that are paralogs in *C. elegans*, *scl*-*24* (upregulated on days 5 and 7) and *scl*-*25* (upregulated on day 5), are predicted to encode peptidase inhibitors (Figure 4D–G). These genes share ~73% homology with human peptidase inhibitor 16 (PI16). We knocked down both *scl*-*24* and *scl*-*25* genes separately to examine the impact on DA neurodegeneration. The knockdown of *scl*-*24* attenuated DA neurodegeneration when exposed to the metabolite compared to EV control RNAi (*p* = 0.0059), indicating that the presence of *scl*-*24* might contribute to metabolite neurodegeneration (Figure 4D). In contrast, the knockdown of the paralog, *scl*-*25*, enhanced DA neurodegeneration, even in the absence of the metabolite (*p* < 0.0001). The addition of the metabolite did not further enhance neurodegeneration, suggesting a possible epistatic relationship (*p* = 0.0004) (Figure 4F). We performed real-time quantitative PCR (RT-qPCR) on *scl*-*24* and *scl*-*25*, to validate these results. Following exposure to *S. ven*, the mRNA level of *scl*-*24* was upregulated on both days 5 and 7 compared to EtAc solvent control, in a pattern that mirrored the RNA-seq data (Figure 4E). RT-qPCR results of *scl*-*25*, a gene that was upregulated only on day 5, also confirmed upregulation following *S. ven* treatment (Figure 4G). These two DEGs are predicted to be localized to the plasma membrane or be secreted molecules and have known or predicted roles associated with immune function. Specifically, functions in inflammation, hypertrophy, and T-cell regulation have been reported [55,56,57,58]. However, an association with neurodegeneration had not been previously assigned to these gene products.

### 3.5. RNAi Reveals Distinct Effects of Phase I Gene Products in S. ven-Induced Neurodegeneration

Our transcriptomic analysis also identified six DEGs encoding Phase I cytochrome P450 enzymes. These monooxygenases have many substrates, including steroids and fatty acids. P450 oxidases also have critical roles in the biotransformation of xenobiotics through oxidative metabolic reactions [59]. We used RNAi to knockdown DEGs encoding four Phase I CPY450 enzymes and then analyzed DA neurodegeneration on day 9 (Figure 5); this analysis included two downregulated (*cyp*-*13A5* and *cyp*-*37B1*) and two upregulated (*cyp*-*34A10* and *cyp*-*35B2*) *cyp* gene products.

Animals that were grown on plates seeded with control RNAi bacteria (EV) that do not target knockdown of any gene reproducibly displayed a significant increase in DA neurodegeneration following exposure to *S. ven*, in comparison to solvent (EtAc) controls (Figure 5B–E). The expression of *cyp*-*13A5* and *cyp*-*37B1* had been characterized as being decreased in response to metabolite exposure on day 5 and day 7, respectively (Table 1 and Table 2). The depletion of these downregulated genes by RNAi resulted in dopaminergic neuroprotection from *S. ven* metabolite exposures when compared to solvent-treated animals targeted for either *cyp*-*13A5* or *cyp*-*37B1* RNAi knockdown (Figure 5B,C). Therefore, these results suggest that the normal activities of these gene products facilitate *S. ven*-induced neurodegeneration. Conversely, among the transcripts of Phase I genes upregulated by *S. ven*, *cyp*-*34A10* enhanced DA neurotoxicity when knocked down by RNAi in a comparison between EtAc- and *S. ven*-treated animals (*p* = 0.0003) (Figure 5D). Moreover, *cyp*-*34A10* RNAi resulted in significant enhancement of neurodegeneration over EV RNAi controls treated with metabolite (*p* = 0.0012). These data suggest that *cyp*-*34A10* normally acts in a protective manner and that the depletion of this gene product exacerbates the neurotoxicity of the *S. ven* metabolite.

The only Phase I enzyme-encoding transcript that was upregulated at both days 5 and 7, *cyp*-*35B2*, also displayed the most enhanced expression in our study at both day 5 (4.57-fold) and day 7 (4.71-fold) (Table 1 and Table 2). We performed RT-qPCR on *cyp*-*35B2* to validate these results. Following exposure to *S. ven*, the mRNA level was upregulated on both days 5 and 7 compared to EtAc solvent control in a pattern that mirrored the RNA-seq data (Figure 5F). RNAi knockdown of *cyp-35B2* resulted in enhanced DA neurodegeneration in the absence of metabolite, suggesting an important role for *cyp*-*35B2* intrinsic to the maintenance of neuronal health (Figure 5E). Interestingly, in worms exposed to the metabolite, RNAi knockdown of *cyp*-*35B2* attenuated DA neurodegeneration compared to EV control RNAi (*p* = 0.0471), thereby indicating that CYP-35B2 contributes to the neurotoxicity of the *S. ven* metabolite (Figure 5E). This caught our attention because most Phase I reactions tend to increase the solubility of compounds and reduce their toxicity. Nevertheless, it is known that some oxidative reactions enhance toxicity through compound bioactivation [22]. In total, we identified differentially regulated CYPs that increased and decreased the neurotoxicity of *S. ven* through their functions in metabolism and bioactivation.

### 3.6. Depletion of Phase II Detoxification Genes Attenuates S. ven-Induced DA Neurodegeneration

In Phase II reactions, substrates are conjugated with a water-soluble group to facilitate excretion. In this manner, products of Phase I-dependent reactions are often coupled to Phase II enzymes to influence the processing of both xenobiotics and endogenous substrates. It is, therefore, significant that among the transcripts that we identified as being upregulated by *S. ven* exposure were *C. elegans ugt*-*22* and *ugt*-*66*, encoding Phase II detoxification enzymes (Table 2). Both DEGs belong to the uridine 5′-diphospho-glucronosylatransferase (UGT) family and are homologous to the human UGT1A1 protein. Using RNAi to knockdown *ugt*-*66* (Figure 5G) and *ugt-22* (Figure 5H) in *C. elegans*, we observed enhanced protection against *S. ven*-induced DA neurodegeneration when compared to EtAc control (*p* = 0.0121 and *p* = 0.0163, respectively). The decreased neurotoxicity observed from the knockdown of these UGTs is notable, as these gene products typically function to reduce compound toxicity, although bioactivation has also been reported [20,21]. RT-qPCR results of *ugt*-*22*, a gene that was upregulated only on day 5, also confirmed upregulation following *S. ven* treatment (Figure 5I). Given these data, it suggests that in *S. ven* metabolite-treated animals, the activity of these gene products contributes to neurodegeneration.

### 3.7. Non-CYP Phase I Gene Products Display Altered S. ven Neurotoxicity following Knockdown

Whereas CYP enzymes are the primary catalysts of the oxidative metabolism of xenobiotics, non-CYP enzymes can also contribute to Phase I metabolism. Notably, we identified three DEGs encoding non-CYP oxidative enzymes in response to the *S. ven* metabolite. These included one upregulated DEG, encoding a flavin-containing monooxygenase (*fmo*-*3*), and two downregulated transcripts encoding alcohol dehydrogenase (*adh*-*1*) and xanthine hydrogenase (*xdh*-*1*) (Table 1 and Table 2). To evaluate the influence of these gene products on metabolite-induced neurodegeneration, we employed RNAi to knockdown these candidates in *C. elegans* (Figure 6A–D).

The *adh*-*1* gene is predicted to encode a gene product enabling NAD^+^-dependent alcohol dehydrogenase activity [60]. In *C. elegans*, this gene was reported to be upregulated in defense against infection with the Gram-positive bacteria *S. aureus* [61] and *M. nematophilum* [62]. In contrast, *adh*-*1* was downregulated on day 5 (Table 1) in our study; *S. ven* is also a Gram-positive bacterium. The RNAi depletion of *adh*-*1* resulted in neuroprotection following exposure to the *S. ven* metabolite when compared to EV solvent controls (*p* = 0.0316). Thus, these data indicate that the normal expression of *adh*-*1* enhances DA neurodegeneration (Figure 6B).

We also examined *xdh*-*1*, for which transcripts were downregulated on day 7 following exposure to *S. ven* (Table 2). We depleted *xdh*-*1* by RNAi to further suppress the expression of this gene product but this did not change the amount of neurodegeneration in comparison to the EV control (Figure 6C). RT-qPCR results of *xdh*-*1* also confirmed downregulation following *S. ven* treatment, consistent with transcriptomic data (Figure 6D). The encoded gene product, xanthine dehydrogenase (XDH-1), belongs to the molybdenum-containing hydroxylases group of enzymes that oxidatively metabolizes purines. In mammals, the XDH protein has been shown to perform a mechanistically distinct function via an interconversion to xanthine oxidase (XO) (Figure 6E). The in vivo transition of XDH to XO results in an increase in ROS, contributing to the production of both O_2_ and H_2_O_2_ through one- and two-electron reduction, leading to oxidative damage [25,63]. This conversion can occur reversibly by conformational change through sulfhydryl oxidation or calcium interaction or irreversibly through proteolytic modification (Figure 6E) [23]. XDH/XO interconversion has been studied widely, including as a defense strategy against infectious pathogens. For example, in mice infected with influenza, the ratio of XO to XDH activity in samples of alveolar lavage fluid increased from ~0.15 to 1.06 [64]. We found it intriguing that RNAi knockdown of *xdh*-*1* did not result in a discernable response (Figure 6C) since the other genes we examined resulted in defined phenotypes following RNAi depletion and exposure to the *S. ven* metabolite (Figure 4 and Figure 5). However, knowing that the XDH-1 gene product can interconvert to XO, we asked if XO enzyme activity was modulated in worms treated with *S. ven* metabolite. We measured XO activity in day 7 adult worms treated with either metabolite or solvent and observed a significant increase in XO activity between the two treatment groups, with *S. ven* treated worms displaying 1.7× the amount of XO activity compared to EtAc (Figure 6F). These results demonstrate a significant shift to XO from XDH following *S. ven* treatment.

### 3.8. The Neurotoxic Effect of the S. ven Metabolite Is Potentiated by Calcium Supplementation

The production of ROS is associated with many processes, including alterations in calcium signaling. Previous studies have identified altered calcium homeostasis playing a central role in the oxidation of XDH and the production of XO [24,25,26]. To determine whether changes in CaCl_2_ levels contribute to neurodegeneration in our model, we added double (2 mM) and triple (3 mM) the amount of CaCl_2_ normally used in nematode growth media without the metabolite. We determined that tripling the CaCl_2_ levels (3 mM) enhanced neurodegeneration (*p* = 0.0098), comparable to levels seen in the metabolite background (Figure 6G). Subsequently, we tested whether the removal of the typical CaCl_2_ supplement from the worm plates (1 mM) would decrease neurodegeneration following metabolite exposure. Strikingly, we discovered that worms exposed to *S. ven* on NGM plates with 0 mM CaCl_2_ supplementation no longer exhibited neurodegeneration (Figure 6H). Importantly, compared to worms treated with the 1 mM CaCl_2_ and the *S. ven* regimen, worms grown on 0 mM CaCl_2_ NGM plates displayed protection against metabolite exposure (*p* < 0.0033) (Figure 6H). To confirm our findings, we took a pharmacological approach and exposed worms to the calcium chelator EGTA, with or without *S. ven*. When *C. elegans* were exposed to 0.5 mM EGTA along with *S. ven*, they no longer displayed DA neurodegeneration from the metabolite (Figure 6I). These results suggest that calcium is important for *S. ven*-induced neurodegeneration. To determine if our calcium data corresponded with XO activity data, we performed two sequential experiments. First, we removed the normal 1 mM CaCl_2_ from the NGM and examined the level of XO in the ex vivo assay (Figure 6J). We found worms grown on 0 mM NGM plates without the normal 1 mM CaCl_2_ supplementation, yet exposed to *S. ven*, no longer displayed significantly higher levels of XO compared to the controls (Figure 6J). Second, we tested whether worms exposed to 0.5 mM EGTA, along with *S. ven*, displayed changes in XO activity levels. Similarly, there was no increased XO activity compared to the control (Figure 6K), complementing both Figure 6J and the neurodegenerative studies. Overall, these results indicate that exposure to *S. ven* metabolite increased *C. elegans* XO activity via increased ROS when CaCl_2_ was provided through the media and that this contributes to neurodegeneration.

## 4. Discussion

While few environmental exposures have been assigned to neurodegenerative disease pathogenesis, the idiopathic nature of these diseases necessitates that less established sources of neurotoxicity be investigated experimentally. Previously, we reported that *S. ven* metabolite(s) caused DA neurodegeneration in *C. elegans* and SH-SY5Y cells [16]. Likewise, exposure to *S. ven* also resulted in mitochondrial dysfunction and oxidative stress [17,18]. A meta-analysis of epidemiological studies reported a higher prevalence of PD in residents of rural areas and/or in individuals who are engaged in farming activities. However, the causative factor(s) remain unidentified and cannot be wholly attributed to factors such as pesticide usage alone [65]. Therefore, we posit that increased human interaction with the soil itself may represent a risk factor, whereby lengthy periods of time and exposure to metabolite(s) from common soil bacteria, such as *S. ven*, lead to neurodegeneration. In such a scenario, susceptible individuals are likely to harbor genetic polymorphisms that render their defenses more vulnerable to a variety of putative stressors. Defining changes in organismal response to *S. ven* represents a pivotal stage in the identification of potential genetic biomarkers of susceptibility through functional annotation of human genomic variants [66].

As a soil-dwelling organism, it is likely that *C. elegans* has honed defense strategies against the toxic insults from soil bacteria, including *S. venezuelae*. In support of this, transcriptional profiling of *C. elegans* following *S. ven* exposure identified a select group of 35 genes modified by this treatment, suggestive of high specificity. Among the principal findings of this study was that *C. elegans xdh*-*1* was downregulated following *S. ven* exposure. This enzyme functions as a non-CYP Phase I enzyme, along with two other DEGs we identified from our transcriptomics study (*fmo*-*3* and *adh*-*1*). It is notable that non-CYP Phase I enzymes are usually non- or less-inducible [67]. In our transcriptomics data set, 44% of the DEGs are associated with metabolic detoxification: ~9% (3/35) encoded non-CYP Phase I enzymes, 20% encoded Phase I or II detoxification gene products, and 15% of the DEGs were broadly associated with innate immunity. Therefore, it is tempting to speculate that these changes in transcriptional regulation reflect an evolutionary adaptation derived from mutual environmental antagonism.

The most upregulated DEG in this study was the gene encoding a Phase I enzyme, *cyp*-*35B2*; it was identified on both days 5 and 7, indicating a sustained transcriptional response. This is not surprising considering that 19 chemicals have previously been shown to regulate the expression of *cyp*-*35B2* in *C. elegans* [68]; these include the common proton pump inhibitor lansoprazole and phenobarbital, which is used to control seizures in humans. The knockdown of *cyp*-*35B2* in the presence of *S. ven* provided evidence that the normal activity of the gene product contributes toward an enhancement of neurotoxicity (Figure 5E). Mechanistically, detoxification usually occurs through an increase in water solubility of an original, pre-metabolically altered compound. Yet, here, the *S. ven* parent compound demonstrates increased toxicity, indicative of its bioactivation through metabolic processing. While these data are interesting to consider, another upregulated Phase I DEG, *cyp*-*34A10,* displayed the opposite phenotype and might normally act in reducing neurotoxicity (Figure 5D). There is scant literature describing the activity of this enzyme in *C. elegans*. However, *cyp*-*34A10* is essential for the metabolism of the diabetes drug tolbutamide [69]. Likewise, a polychlorinated biphenyl compound, PCB52 (2,5,2’,5’-tetrachlorobiphenyl), was also shown to be an effector of *cyp*-*34A10* in *C. elegans* [70].

Two other Phase I DEGs we analyzed were downregulated in our transcriptomics analysis (*cyp*-*13A5* and *cyp*-*37B1*). When we further reduced their function and added the metabolite, neuroprotection resulted (Figure 5B,C), again suggesting that CYP-35B2 bioactivated the parent compound through Phase I enzymatic processing. Interestingly, PCB52 not only upregulates *cyp*-*34A10*, but it also upregulates *cyp*-*37B1* [70]. Thus, it would be interesting to know if CYP-34A10 and CYP-37B1 function in a regulatory network to oxidatively metabolize xenobiotic substances. Similarly, another one of our DEGs, *cyp*-*13A5*, has been previously reported to function with multiple CYPs, perhaps suggestive of an extensive regulatory network [71,72].

We identified two upregulated DEGs encoding Phase II enzymes (UGT-22 and UGT-66); they are both within the UDP-glucuronosyltransferase family. The knockdown of these targets in our *C. elegans* model attenuated the neurodegeneration resulting from *S. ven* metabolite treatment (Figure 5). These data suggest that the *S. ven* parent compound becomes more toxic through bioactivation via the activity of both Phase I and Phase II enzymes. While these genes encode family members of a main class of Phase II enzymes, there are several other Phase II detoxification enzyme families that were not represented in our dataset. Considering that DA neurodegeneration is most evident beginning at day 9, and we collected RNA on days 5 and 7 for our transcriptomics analyses, it is quite possible that we did not capture the range of Phase II enzymes associated with *S. ven* exposure.

We were initially puzzled by the identification of three DEGs encoding non-CYP Phase I enzymes, considering that these enzymes are typically less inducible. However, since most of the Phase I and II enzymes bioactivated *S. ven*, additional gene regulation to potentially reduce this toxicity represents a logical cellular response. For example, we determined that *S. ven* metabolite exposure downregulated xanthine dehydrogenase (*xdh*-*1*) gene expression, with a concomitant increase in XO enzyme activity in *C. elegans* (Figure 6). Physiologically, both enzymes can participate in the detoxification of xenobiotics and endogenous compounds but also contribute to oxidative stress environments [73]. This suggests that in the presence of *S. ven*, when CYP Phase I enzyme metabolism bioactivates the metabolite, it could lead to increasing levels of ROS and shift XDH to XO. Of the two forms, XO is primarily associated with ROS generation. If the production of hydrogen peroxide and molecular oxygen overwhelm organism scavenging systems, damage to intracellular targets and pathways will occur [24], and XO enzyme levels will be higher in diseased or damaged conditions [74].

Taken together with previously reported interactions between XO and calcium [24,25,26], our data support a mechanism whereby the *S. ven* metabolite stress response is dependent on calcium. Specifically, 1 mM CaCl_2_ supplementation was necessary for *S. ven* neurodegenerative activity (Figure 6H), while 3 mM CaCl_2_ was sufficient for inducing neurodegeneration even in the absence of the *S. ven* metabolite (Figure 6G vs. Figure 6H). We measured the ratio of XO to XDH and discovered that *S. ven* increased XO levels in worms exposed to *S. ven* compared to worms exposed to solvent control; moreover, the significant XO increase occurred when there was also 1 mM CaCl_2_ in the media but not when there was 0 mM CaCl_2_ in the media (Figure 6F vs. Figure 6J). Markedly, 0 mM CaCl_2_ in the media, along with *S. ven* exposure, ameliorated neurodegeneration (Figure 6H). *E. coli,* the food source of *C. elegans*, have their own intracellular calcium stores [75]. Therefore, data interpretation can be complicated when considering the endogenous calcium within *E. coli* that *C. elegans* are exposed to. Nonetheless, when the calcium chelator EGTA was added to the NGM media, we found that the XO level became non-significantly different between solvent control and *S. ven* treated animals (Figure 6K). Similarly, EGTA treatment ameliorated *S. ven* neurotoxicity (Figure 6I)_._ These data are reminiscent of previous data where it was shown that reducing calcium stores decreased the production of XO [76]. Here, we show that two stimulants, *S. ven* and calcium, can enhance neurodegeneration through XO production. It is interesting to note that *xdh*-*1* was a downregulated DEG in our transcriptomics dataset. We hypothesize that the downregulation of *xdh*-*1* in response to metabolite exposure is a defense mechanism that delimits the pool of XDH-1 available for interconversion to XO. This will then also limit ROS production.

Notably, the dysregulation of calcium homeostasis is associated with neurological disorders such as PD [77,78,79]. XDH is a key enzyme in the catabolism of purine nucleotides to uric acid. Consistent with our data, the upstream regulator of XDH, inosine, is protective against neurodegeneration in both in vitro and in vivo PD models [80,81]. A small clinical trial was initiated to test inosine as a therapeutic for PD patients. However, it was halted because the clinical progression of PD was not slowed by the treatment with oral inosine [82].

It is notable that Phase III molecules were not identified in our transcriptomics analysis. They have a critical role in the final step of compound excretion and include solute carriers, P-glycoproteins (*pgp*), ATP-binding cassettes, and ion channels. In retrospect, if we had isolated RNA at a later time point (i.e., day 8 or 9), we might have identified Phase III detoxification DEGs. However, it is interesting that no early Phase III DEGs were captured on day 7. We cannot rule out that the metabolite may never be fully excreted from *C. elegans*, and this leads to overaccumulation and cellular damage. Another possibility is that Phase III transporters do not need to be upregulated to facilitate the excretion of the metabolite.

While we did not identify DEGs putatively associated with metabolite removal, ~30% of the DEGs displayed a connection to the DAF-16 transcription factor through genetics, microarray analysis, lifespan studies, or bioinformatic analyses (Table 4). Since DAF-16 is a major transcriptional regulator of stress response [83], these data support a hypothesis where a stress response signature is activated that includes DAF-16 when *C. elegans* encounters *S. ven*. This premise is supported by prior data where we discovered that *S. ven* exposures led to significant relocalization of DAF-16 from the cytosol to the nucleus [17]. We determined that 60% of our DEGs contained either a *daf*-*16*-binding element (DBE) or a *daf*-*16*-associated element (DAE) from a bioinformatics analysis. This suggests a sustained role for DAF-16 following exposure to the *S. ven* metabolite. Many of the genes with the DAF-16 binding elements were associated with the GO terms “detoxification” and “stress response” according to WormCat analysis. Because DAF-16 can modify expression levels both directly and indirectly through binding motif sequences (DBE and DAE, respectively), we asked whether there was a transcriptional requirement for DAF-16 following metabolite exposure, using RT-qPCR with four exemplar DEGs. We determined that DAF-16 positively regulated the expression of *cyp-35B1,* and we speculate that DAF-16 negatively regulated *cyp*-*34A10*. We also examined the transcription factor PQM-1 for a role in modulating gene expression in the presence of *S. ven* metabolite. We identified a requirement for PQM-1 in *fmo*-*3* gene expression. Occasionally, PQM-1 and DAF-16 co-regulate targets, as has been previously described for a small subset of DAF-16 targets that contain both binding elements [49]. We identified that *set-18* was transcriptionally regulated by both PQM-1 and DAF-16 in the presence of *S. ven*.

## 5. Conclusions

In total, we hypothesize that as part of a strategic defense mechanism in the soil, the nematode has honed a generalized mechanism by which a transcriptional response is mounted to disable toxic substances through multiple up- and down-regulated enzymatic reactions. It is widely believed that a combination of gene-by-environment interactions might contribute to the onset and/or progression of neurodegeneration. These environmental exposures could include toxins from natural sources that induce cellular oxidative stress. Chronic, and perhaps, low-level exposure to environmentally produced compounds exacerbates neurodegeneration but might also provide a window into therapeutic benefit. For example, in a companion paper in preparation, we chronically exposed *C. elegans* to a lower concentration of *S. ven* metabolite (20× herein, 5× in the subsequent study). With the lower (5×) concentration, we identified a protective, hormetic response whereby the lifespan of *C. elegans* is extended and did not cause increased DA neurodegeneration. In this follow-up study (Thies and Caldwell in preparation), we assessed many of the same DEGs. As such, the *C. elegans* defense machinery acts as a double-edged sword, dependent on dose. This hormetic response represents a putative entrée toward defining therapeutic benefits for oxidative stress-related cellular mechanisms. The targets revealed through the transcriptomic analysis and functional evaluation herein represent an opportunity to mechanistically discern common environmental contributors to oxidative stress-induced neurodegeneration. The prevalence and societal burden of PD demand that greater consideration and resources be devoted toward the unseen but imminent threat that microbial sources of neurotoxins potentially represent.

## Figures and Tables

**Figure 1 cells-12-01170-f001:**
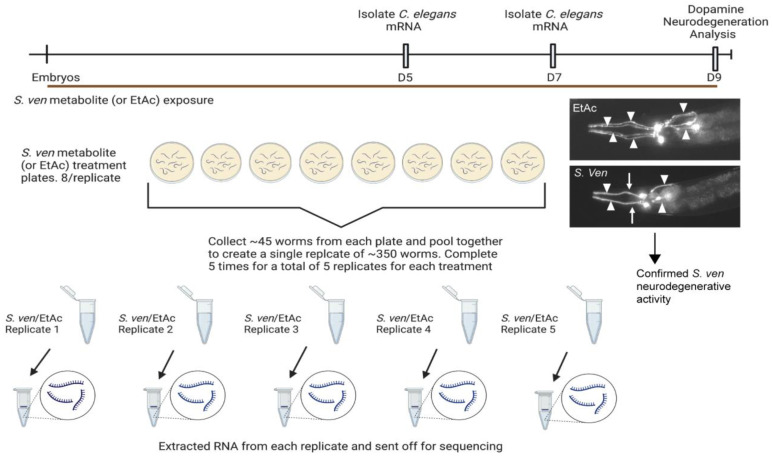
Schematic diagram of *C. elegans* in preparation for RNA sequencing analysis following *S. ven* metabolite exposure. Worms were grown in the presence of chronic *S. ven* metabolite or EtAc exposure for the duration of the experiment. On Days 5 and 7 of adulthood, mRNA was extracted from whole *C. elegans* animals. On each day, ~350 worms were collected from each treatment to comprise a single replicate. mRNA was extracted and subsequently sequenced. To ensure *S. ven* metabolite contained active neurodegenerative capacity, additional worms from the exposure were grown until day 9 of adulthood and then analyzed for dopaminergic (DA) neurodegeneration. These worms expressed GFP in the DA neurons (P*_dat-1_*::GFP), which readily allows for phenotypic analysis of neurodegenerative phenotypes. Representative images of the anterior region of *C. elegans* expressing P*_dat-1_*::GFP are displayed in this figure. There are six DA neurons in the anterior region of a healthy worm. Here, DA neurons on day 9 of adulthood when exposed to EtAc (top) or *S. ven* metabolite (bottom) are shown. Arrowheads indicate intact neurons, whereas arrows indicate degenerated neurons. Created with BioRender.com (accessed on 9 September 2022).

**Figure 2 cells-12-01170-f002:**
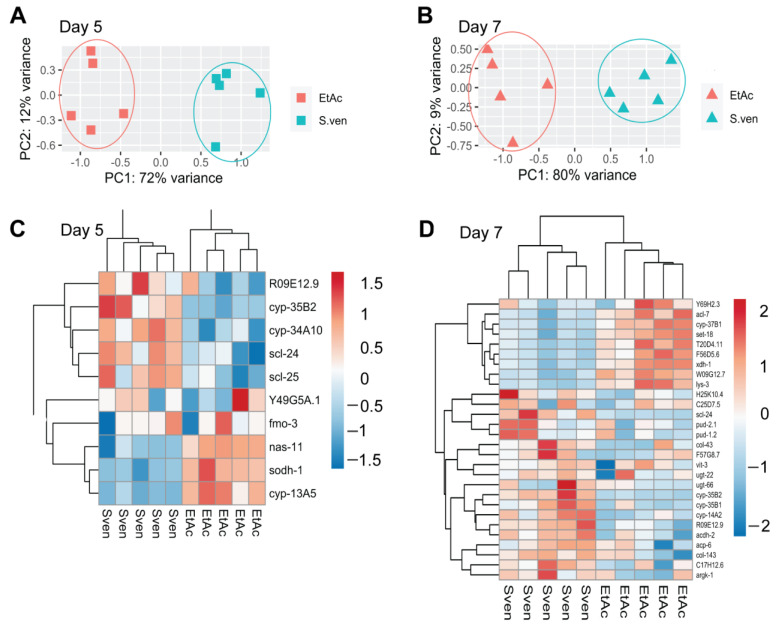
PCA plots and heatmaps of five replicates of differentially expressed genes as determined by the adjusted *p*-value. (**A**,**B**) Principal component analysis (PCA) showing relative distance among samples at day 5 (**A**) and day 7 (**B**). PC1 represents the variance between treatments, and PC2 represents the variance between replicates. Replicates from day 5 displayed a 72% variance between EtAc and the *S. ven* metabolite treatments, while the variance between replicates accounted for only 12% (**A**). Replicates from day 7 displayed an 80% variance between EtAc and *S. ven* metabolite treatments, while the variance between replicates accounted for only 9% (**B**). Red represents EtAc treatment, and blue represents *S. ven* treatment in (**A**,**B**). (**C**,**D**) Heatmap of Z-score normalized gene expression displaying significant DEGs at day 5 (*p*adj < 0.054) (**C**) and day 7 (*p*adj < 0.057) (**D**).

**Figure 3 cells-12-01170-f003:**
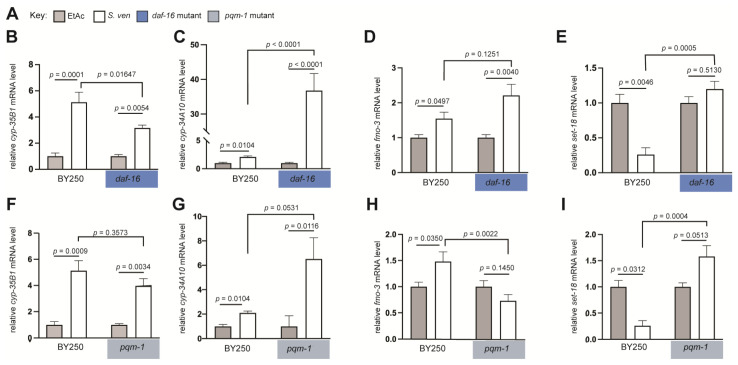
DEGs modulated by *daf*-*16* and *pqm*-*1* in response to *S. ven* metabolite. Quantification of relative mRNA expression levels of genes containing DAF-16 regulatory motifs was measured in *daf*-*16* (dark blue background) or *pqm*-*1* (light blue background) mutants following exposure to *S. ven* metabolite or EtAc treatments. (**A**) Key indicating gray bars represent EtAc exposure, and white bars represent *S. ven* exposure; dark blue boxes represent *daf*-*16* mutants and light blue boxes represent *pqm*-*1* mutants. (**B**) *cyp*-*35B1* mRNA expression was measured in day 7 BY250 and *daf-16* mutant adult worms. (**C**) *cyp*-*34A10* mRNA expression was measured in day 7 BY250 and *daf-16* mutant adult worms. (**D**) *fmo*-*3* mRNA expression was measured in day 7 BY250 and *daf-16* mutant adult worms. (**E**) *set*-*18* mRNA expression was measured in day 7 BY250 and *daf*-*16* mutant adult worms. (**F**) *cyp*-*35B1* mRNA expression was measured in day 7 BY250 and *pqm*-*1* mutant adult worms. (**G**) *cyp*-*34A10* mRNA expression was measured in day 7 BY250 and *pqm*-*1* mutant adult worms. (**H**) *fmo*-*3* mRNA expression was measured in day 7 BY250 and *pqm*-*1* mutant adult worms. (**I**) *set*-*18* mRNA expression was measured in day 7 BY250 and *pqm*-*1* mutant adult worms. In all graphs, the normalized mean fold change of all biological replicates is relative to the EtAc control for each strain. Values represent mean +/− SEM of 3–4 biological replicates with three technical replicates; at least 100 animals were used for each replicate. Significance was obtained via two-way ANOVA with Tukey’s post hoc and is represented using exact *p*-values (n = 120 per replicate, N = 3–4 biological replicates).

**Figure 4 cells-12-01170-f004:**
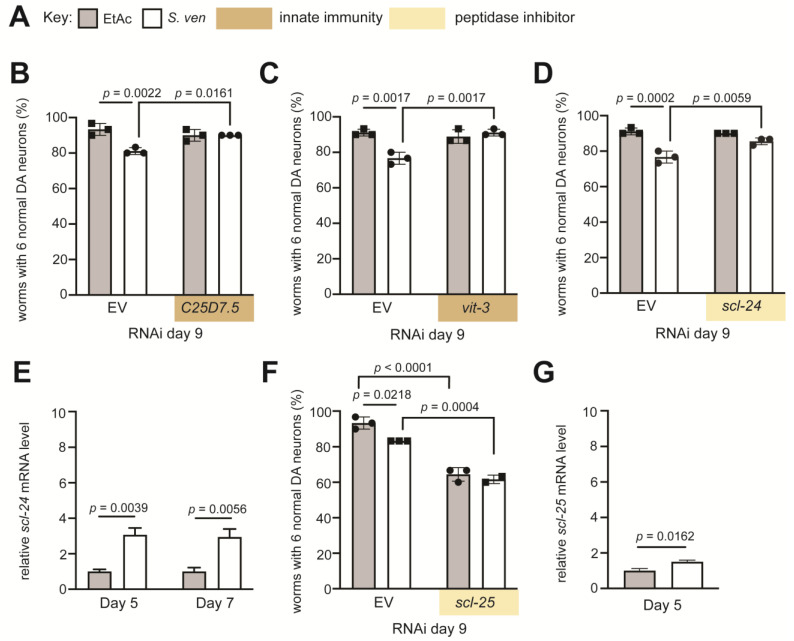
RNAi knockdown of innate immunity genes attenuates *S. ven*-induced DA neurodegeneration in *C. elegans*. (**A**) Key denoting gray bars represent EtAc, white bars represent *S. ven* treatment, tan represents genes associated with innate immunity, and light yellow indicates genes associated with peptidase inhibitor activity. (**B**–**D**,**F**) DA neurodegeneration analysis in P*_dat-1_*::GFP day 9 adult animals following RNAi of genes associated with either innate immunity (**B**,**C**) or peptidase inhibitor activity (**D**,**F**). (**B**) *C25D7.5* and (**C**) *vit*-*3* results in decreased DA neurodegeneration following *S. ven* metabolite exposure. (**D**) Loss of peptidase inhibitor *scl-24* results in decreased DA neurodegeneration in animals exposed to *S. ven* metabolite. (**F**) Loss of peptidase inhibitor *scl*-*25* results in enhanced DA neurodegeneration in animals following *S. ven* exposure. Neurodegenerative represented as mean +/− SD. Data analyzed by two-way ANOVA with Tukey’s post hoc test (n = 30 animals/rep; 3 biological replicates). (**E**,**G**) Quantification of mRNA gene expression levels of peptidase inhibitors (**E**) *scl*-*24* and (**G**) *scl*-*25* on adult day 5 or 7 worms following exposure to either EtAc or *S. ven* metabolite. mRNA data analyzed by one-tailed Student’s *t*-test and presented as mean +/− SEM of three biological replicates, with three technical replicates each; at least 100 animals were used for each replicate. All RT-qPCR data normalized to EtAc control. Significance is represented by exact *p*-values.

**Figure 5 cells-12-01170-f005:**
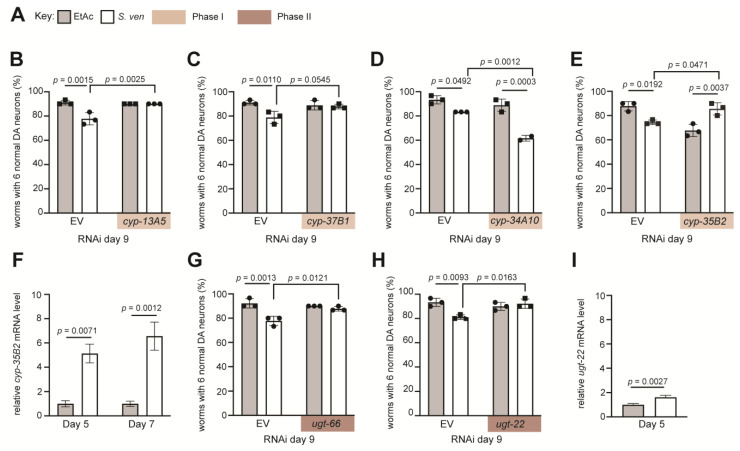
Phase I and phase II detoxification genes modulate DA neurodegeneration in *C. elegans* following *S. ven* exposure. (**A**) Key denoting gray bars represent EtAc, white bars represent *S. ven* treatment, peach indicates Phase I genes and brown indicates Phase II genes. (**B**–**E**) Phase I detoxification genes were knocked down in P*_dat-1_*::GFP adult, day 9 animals following chronic exposure to EtAc or *S. ven* metabolite. Knockdown of (**B**) *cyp*-*13A5*, (**C**) *cyp*-*37B1* and (**E**) *cyp*-*35B2* results in decreased DA neurodegeneration following *S. ven* metabolite exposure compared to the EV *S. ven* metabolite control. Knockdown of *cyp*-*34A10* enhanced DA neurodegeneration in an *S. ven* metabolite background alone and in comparison, to (**D**) the EV *S. ven* metabolite control. (**G**,**H**) Knockdown of Phase II enzymes (**G**) *ugt*-*22* and (**H**) *ugt*-*66* results in decreased neurodegeneration when exposed to *S. ven* metabolite compared to the EV *S. ven* metabolite control. Neurodegeneration data represented as mean +/− SD. n = 30 animals per replicate, three independent replicates per treatment; two-way ANOVA with Tukey’s post hoc test for multiple comparisons. (**F**,**I**) Quantification of mRNA gene expression levels of gene expression of (**F**) *cyp*-*35B2* and (**I**) *ugt-22* on adult worms following exposure to either EtAc or *S. ven* metabolite on the day(s) indicated. mRNA data analyzed by one-tailed Student’s *t*-test and presented as mean +/− SEM of three biological replicates, with three technical replicates each; at least 100 animals were used for each replicate. All RT-qPCR data normalized to EtAc control. Significance is represented by using exact *p*-values.

**Figure 6 cells-12-01170-f006:**
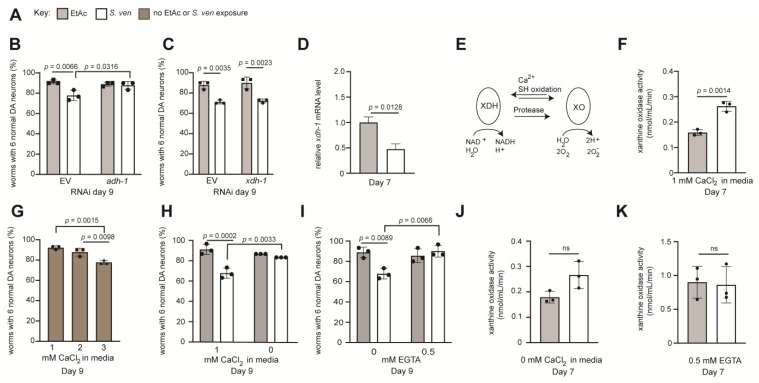
*S. ven* metabolite induces xanthine oxidase activity dependent on the presence of calcium. (**A**) Key in which the gray bars represent EtAc treatment, white bars are *S. ven* treatment, and light brown bars have no treatment. (**B**) *C. elegans* DA neurodegeneration on adult day 9 animals following RNAi of *adh*-*1* in the presence of EtAc or *S. ven* metabolite treatment. Loss of *adh*-*1* resulted in decreased neurodegeneration in animals exposed to *S. ven* metabolite alone and compared to the *S. ven* metabolite EV control. (**C**) *C. elegans* DA neurodegeneration in adult day 9 animals (P*_dat-1_*::GFP) following RNAi of *xdh*-*1* in the presence of EtAc or *S. ven* metabolite treatment. Loss of *xdh-1* had no impact on DA neurodegeneration in animals exposed to *S. ven* metabolite (**D**) Quantification of *xdh*-*1* mRNA gene expression levels in adult day 7 animals following exposure to EtAc or *S. ven* metabolite. (**E**) Model depicting interconversion of XDH to XO. The accumulation of XO can occur through an interaction with calcium; it increases ROS. (**F**,**J**,**K**) Ex vivo xanthine oxidase activity assays. (**F**) Xanthine oxidase activity was measured in day 7 adult worms following chronic exposure to EtAc or *S. ven* metabolite treatment when the normal amount of calcium (1 mM) was added to the worm media. (**G**) *C. elegans* DA neurodegeneration in adult day 9 animals (P*_dat-1_*::GFP) following exposure to increasing amounts of CaCl_2_ (1, 2 or 3 mM) within the NGM media. 3 mM CaCl_2_ resulted in enhanced neurodegeneration. (**H**) Adult day 9 animals grown on plates without CaCl_2_ (0 mM CaCl_2_) within the NGM media displayed reduced DA neurodegeneration following chronic exposure to *S. ven* metabolite compared to the 1 mM *S. ven* control. (**I**) *C. elegans* were exposed to 0 or 0.5 mM EGTA, along with *S. ven*. Neuroprotection from *S. ven* was observed when worms were also exposed to the calcium chelator, EGTA. (**J**) Xanthine oxidase activity was measured in day 7 adults chronically exposed to EtAc or *S. ven* metabolite when worms were grown on plates when CaCl_2_ was omitted from the NGM media. (**K**) Xanthine oxidase activity was measured in day 7 adults chronically exposure to EtAc or *S. ven* metabolite when worms were grown on plates supplemented with 0.5 mM EGTA in addition to EtAc or *S. ven* metabolite. For all neurodegeneration experiments, data are represented as mean +/− SD; n = 30 animals per replicate, three independent replicates per treatment; one-way ANOVA or two-way ANOVA with Tukey’s post hoc test for multiple comparisons. Xanthine oxidase activity was analyzed with Student’s *t*-test and presented as mean +/− SD; at least 120 animals were used for each replicate, 3 biological replicates. The RT-qPCR data were normalized to EtAc control. Significance is represented by using exact *p*-values. qPCR mRNA data analyzed by Student’s *t*-test and presented as mean +/−SEM of three biological replicates, with three technical replicates each; at least 100 animals were used for each replicate. ns = not significant.

**Table 1 cells-12-01170-t001:** Functional Annotation of Day 5 DEGs.

Wormbase ID	Gene	Human Homolog(s)	Function	Fold Change	*p*adj
Stress Response					
WBGene00019471	*cyp*-*35B2*	CYP2C8	Detoxification	4.57	3.08 × 10^−5^
WBGene00044514	*R09E12.9*	n/a	Responsive to multiple stresses	1.78	5.43 × 10^−2^
WBGene00015045	*cyp*-*34A10*	CYP2A6	Detoxification	1.71	2.20 × 10^−2^
WBGene00011672	*cyp*-*13A5*	CYP3A5	Detoxification	(−)0.50	2.02 × 10^−2^
Metabolism					
WBGene00001478	*fmo*-*3*	FMO5	Monooxygenase	0.83	7.64 × 10^−3^
WBGene00010790	*adh*-*1*	ADH4	Mitochondria	(−)0.48	5.21 × 10^−2^
Proteolysis					
WBGene00011841	*scl*-*25*	PI16	Inhibitor (Peptidase)	2.43	4.69 × 10^−3^
WBGene00008575	*scl-24*	PI16	Inhibitor (Peptidase)	1.96	9.72 × 10^−9^
WBGene00021731	*Y49G5A.1*	WFDC8	Inhibitor (Cysteine)	0.85	1.45 × 10^−2^
WBGene00003530	*nas*-*11*	TLL1	Metallopeptidase	0.28	2.02 × 10^−2^

**Table 2 cells-12-01170-t002:** Functional Annotation of Day 7 DEGs.

Wormbase ID	Gene	Human Homolog(s)	Function	Fold Change	*p*adj
Stress Response					
WBGene00019471	*cyp*-*35B2*	CYP2C8	Detoxification	4.71	9.41 × 10^−10^
WBGene00044514	*R09E12.9*	n/a	Responsive to multiple stresses	2.02	2.35 × 10^−3^
WBGene00019472	*cyp*-*35B1*	CYP2C8	Detoxification	1.62	7.75 × 10^−4^
WBGene00010706	*cyp*-*14A2*	CYP2E1	Detoxification	1.57	9.36 × 10^−6^
WBGene00015932	*C17H12.6*	n/a	Innate Immunity (Pathogen)	1.47	5.66 × 10^−2^
WBGene00007455	*ugt*-*22*	UGT1A10	Detoxification	0.73	2.41 × 10^−2^
WBGene00016013	*ugt*-*66*	UGT1A3	Detoxification	0.73	3.45 × 10^−2^
WBGene00007717	*C25D7.5*	MBLAC1	Innate Immunity (Pathogen)	0.68	4.87 × 10^−2^
WBGene00010150	*F56D5.6*	n/a	Responsive to multiple stresses	(−)0.57	2.07 × 10^−2^
WBGene00020617	*T20D4.11*	n/a	Responsive to multiple stresses	(−)0.59	2.68 × 10^−2^
WBGene00009226	*cyp*-*37B1*	CYP4V2	Detoxification	(−)0.80	2.38 × 10^−2^
WBGene00021121	*W09G12.7*	n/a	Responsive to multiple stresses	(−0.92	1.29 × 10^−4^
Protein Modification					
WBGene00044070	*set*-*18*	SMYD3	Methyltransferase	(−)0.71	2.38 × 10^−2^
Metabolism					
WBGene00015894	*acdh*-*2*	ACADSB	Lipid; beta oxidation	1.26	5.66 × 10^−2^
WBGene00009706	*argk*-*1*	CKMT1B, CKMT2, CKB	Creatine kinase	0.87	5.66 × 10^−2^
WBGene00006927	*vit*-*3*	FCGBP	Lipid transport	0.80	3.24 × 10^−3^
WBGene00010083	*xdh*-*1*	XDH, AOX1	Oxidoreductase	(−)0.49	3.24 × 10^−3^
WBGene00012911	*acl*-*7*	GNPAT	Fatty acid metabolism	(−)0.55	2.68 × 10^−2^
Lysosome					
WBGene00022245	*acp*-*6*	ACPP	Acid phosphatase	0.78	2.38 × 10^−2^
Proteolysis					
WBGene00008575	*scl*-*24*	PI16	Peptidase Inhibitor	2.02	1.64 × 10^−8^
WBGene00003092	*lys*-*3*	n/a	Lysozyme	(−)0.99	3.24 × 10^−3^
Extracellular Material					
WBGene00000620	*col*-*43*	COL21A1	Collagen	1.42	3.27 × 10^−2^
WBGene00000716	*col*-*143*	COL6A5	Collagen	0.80	2.38 × 10^−2^
Unassigned					
WBGene00010413	*H25K10.4*	n/a	Unassigned	1.30	8.46 × 10^−3^
WBGene00010216	*F57G8.7*	n/a	Unassigned	1.10	5.66 × 10^−2^
WBGene00017490	*pud*-*2.1*	n/a	Unassigned	0.95	2.07 × 10^−2^
WBGene00021236	*pud*-*1.2*	n/a	Unassigned	0.94	2.38 × 10^−2^
WBGene00013481	*Y69H2.3*	n/a	Unassigned	0.44	3.62 × 10^−3^

**Table 3 cells-12-01170-t003:** WormCat Enrichment Results for all DEGs combined.

Function	Category	Number of Genes in Category
Biological Process	Proteolysis	5
Molecular Function	↳ Inhibitor	3
Cellular Process	↳ Peptidase Inhibitor 16	2
Biological Process	Stress Response	16
Molecular Function	↳ Detoxification	8
Cellular Process	↳ Cytochrome	6

**Table 4 cells-12-01170-t004:** DEGs with evidence for DAF-16 regulation. Shaded rows indicate day 5 and 7 DEGs (light and dark, respectively).

*C. elegans* Gene Name	Gene Product Description	DAF-16 Regulatory Motifs; 1 Kb Upstream of ATG	Citation
*cyp*-*34A10*	Cytochrome P450	1 DAE	this study
*fmo*-*3*	Dimethylaniline monooxygenase	1 DAE	[42]
*adh*-*1*	Alcohol dehydrogenase	1 DAE	[37,42]
*cyp-13A5*	Cytochrome P450		[42]
*cyp-14A2*	Cytochrome P450	1 DAE; 1 DBE	
*ugt-66*	UDP-glucuronosyltransferase	1 DAE	[42]
*acdh-2*	Acyl-CoA dehydrogenase	1 DBE	[42]; this study
*acl-7*	Dihydroxyacetone phosphate acyltransferase	1 DAE	[42]
*acp-6*	Prostatic acid phosphatase	1 DAE	
*col-43*	Collagen	1 DAE	[42]
*F57G8.7*	Unknown	2 DAE	
*T20D4.11*	Unknown	1 DBE	
*W09G12.7*	Unknown	1 DAE	
*Y69H2.3*	Unknown	1 DBE	
*pud-1.2*	Upregulated in *daf*-*2* mutants	2 DAE	
*ud-2.1*	Upregulated in *daf*-*2* mutants	1 DAE	
*cyp-35B1*	Cytochrome P450		[41]; this study
*cyp-37B1*	Cytochrome P450		[42]
*ugt-22*	UDP-glucuronosyltransferase		[42]
*C25D7.5*	Unknown		[37]
*set-18*	SET domain containing		[42,43]; this study
*F56D5.6*	Unknown		[42]
*lys-3*	lysozyme		[42]

## Data Availability

All data are contained within the present manuscript with the exception of the transcriptomic analysis data. These data are available at: Gene Expression Omnibus (GEO) Dataset Series: GSE217300.

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
