# Peer review of "Xanthine Dehydrogenase Is a Modulator of Dopaminergic Neurodegeneration in Response to Bacterial Metabolite Exposure in C. elegans"

_cells, 2023, doi:10.3390/cells12081170_

Round 1

Reviewer 1 Report

Summary

In this article the authors follow up on recent results indicating metabolite(s) of Streptomyces venzuelae, can cause degeneration of dopaminergic neurons in C. elegans. Here, using extracted conditioned media, the authors conduct transcriptomic, RNAi and media supplementation experiments to further define potential mechanisms that underlie metabolite-induced neurodegeneration. Overall I found very few issues with the approach, experimental design or results. The current design, e.g., timing of analysis, etc., is supported by previous results. The statistical analyses (save one question below) are appropriate, and well described. However, I am struggling, a bit, with the calcium-supplementation data (see below). Overall, I support publication of this manuscript, with additional discussion by the authors. This is a really nice paper and the authors are to be commended on the study.

Major concerns

The neurodegenerative phenotype supports a role for Ca2+ in enhancing degeneration. Where I am having difficulty is the concentration differences. It is my understanding that completely scrubbing divalent cations from media solutions and/or bacterial cells is extremely difficult. I have found a few different papers that suggest E. coli grown in LB have an average calcium content of ~0.5-1.5 mM alone. If this is correct, then the food available to the animals varies in calcium content by the same variance in media supplementation that has an effect on the worms.

At the risk of dithering, on the one hand, the data are clear. Additional calcium in the media enhances neurodegeneration, chelators block degeneration. On the other, it is unclear how much the total media + food varies from experiment to experiment. Perhaps I am overthinking this, but it would be helpful if the authors could comment on this. Perhaps it’s the use of the term “exogenous CaCl2” since I believe the worms are still getting significant Ca2+ from their diet, and it’s not clear that “free” calcium in the media would be bioavailable.

Finally, in line with this overall concern, I think that to make the argument that calcium levels induce the production of the XO enzyme, then I would like to see the XO activity in the EGTA-supplemented animals, presumably in that cohort, the metabolite would not result in increased levels of activity.

Minor concerns

In the treatment by RNAi experiments, the authors indicate the use of an ANOVA. Did the two-way ANOVA include an interaction term? It was unclear from the methodology described in 2.13.

Notes

In the materials and methods, there are several places where it seems that the units are not correct, for example, did the authors add 60 milliliters or microliters of extract to the plates? This is throughout the methodology section, so I am pretty sure it’s a typography conversion error, but please double check.

Please define DBE and DBA in methods 2.8

In section 3.3, I realize the authors are going DEG by DEG, but it would have been more intuitive (to me) to discus genes that were daf-16 dependent, then those that were pqm-1 dependent. That is, grouping them each as a set. I don’t feel strongly about this, but it felt a bit uneven as I read it. One thought I had was perhaps they could be clustered (or color coded) within Figure 3 to better orient the readers to the conclusion of the regulatory roles for the 2 transcription factors.

Author Response

Reviewer #1: 
Sufficient background and relevant references? Yes 

References relevant to research? Yes 
Research design appropriate? Yes 
Methods described? Yes 
Results clearly presented? Yes 

Comments: In this article the authors follow up on recent results indicating metabolite(s) of Streptomyces venzuelae, can cause degeneration of dopaminergic neurons in C. elegans. Here, using extracted conditioned media, the authors conduct transcriptomic, RNAi and media supplementation experiments to further define potential mechanisms that underlie metabolite-induced neurodegeneration. Overall I found very few issues with the approach, experimental design or results. The current design, e.g., timing of analysis, etc., is supported by previous results. The statistical analyses (save one question below) are appropriate, and well described. However, I am struggling, a bit, with the calcium-supplementation data (see below). Overall, I support publication of this manuscript, with additional discussion by the authors. This is a really nice paper and the authors are to be commended on the study.

We appreciate your comments and suggestions for our manuscript; we are addressing your concerns.

[Major concerns]

The neurodegenerative phenotype supports a role for Ca2+ in enhancing degeneration. Where I am having difficulty is the concentration differences. It is my understanding that completely scrubbing divalent cations from media solutions and/or bacterial cells is extremely difficult. I have found a few different papers that suggest E. coli grown in LB have an average calcium content of ~0.5-1.5 mM alone. If this is correct, then the food available to the animals varies in calcium content by the same variance in media supplementation that has an effect on the worms.At the risk of dithering, on the one hand, the data are clear. Additional calcium in the media enhances neurodegeneration, chelators block degeneration. On the other, it is unclear how much the total media + food varies from experiment to experiment. Perhaps I am overthinking this, but it would be helpful if the authors could comment on this. Perhaps it’s the use of the term “exogenous CaCl2” since I believe the worms are still getting significant Ca2+ from their diet, and it’s not clear that “free” calcium in the media would be bioavailable.

Thank you for the thoughtful comment. You are correct as the worms are still able to obtain calcium from their E. coli diet and that this is also an exogenous Ca2+ source. We added this caveat to our Discussion section. Moreover, to clarify our writing, we removed the word “exogenous” from the Results section and Abstract.

Additionally, for all experiments performed within a certain set, the same E. coli liquid culture was used, and the same worm media plates were used, so presumably, there shouldn’t be any variation between the same cohort of materials. This was done to try and control as many variables as possible.

Finally, in line with this overall concern, I think that to make the argument that calcium levels induce the production of the XO enzyme, then I would like to see the XO activity in the EGTA-supplemented animals, presumably in that cohort, the metabolite would not result in increased levels of activity. Thank you for this experimental suggestion. We performed it, as suggested (Figure 6K). Indeed, if you compare 6K (EGTA supplementation) with 6J, which has 0 mM CaCl2 in the media, you will observe that they both indicate that XO activity is non-significantly different in S. ven vs. control. However, the data are “cleaner” in the EGTA experiment, thus giving us more confidence that the presence of Ca2+ with S. ven leads to a conversion of XDH to XO (and neurotoxicity).

[Minor concerns]

In the treatment by RNAi experiments, the authors indicate the use of an ANOVA. Did the two-way ANOVA include an interaction term? It was unclear from the methodology described in 2.13. A full model fit was applied. This has been corrected.

[Notes]

In the materials and methods, there are several places where it seems that the units are not correct, for example, did the authors add 60 milliliters or microliters of extract to the plates? This is throughout the methodology section, so I am pretty sure it’s a typography conversion error, but please double check. This was a typography error and has been corrected in the text.

Please define DBE and DBA in methods 2.8 This has been corrected.

In section 3.3, I realize the authors are going DEG by DEG, but it would have been more intuitive (to me) to discus genes that were daf-16 dependent, then those that were pqm-1 dependent. That is, grouping them each as a set. I don’t feel strongly about this, but it felt a bit uneven as I read it. One thought I had was perhaps they could be clustered (or color coded) within Figure 3 to better orient the readers to the conclusion of the regulatory roles for the 2 transcription factors. Thank you for the suggestion for this section of the paper. We have reorganized the figure and text to represent a more cohesive discussion of the tested genes. We have edited the text and figure to discuss the daf-16 dependent genes first, and then the genes that are pqm-1 dependent. Both transcriptional regulators are associated with a different color in the figure.

Reviewer 2 Report

The manuscript titled "Xanthine dehydrogenase is a modulator of dopaminergic  neurodegeneration in response to bacterial metabolite exposure in C. elegans"  by Thies et al is a  dense  and rich paper in which there are a large amount of  data. However, the manuscript results difficult to read. In my opinion it is necessary a more systematic organization of  some  paragraphs of the results. The subsection of the results 3.1 contains informations that can be inserted in Materials and Methods sections (the first part related to fig 1). Moreover the results are reported in a prolix way with comments and references that may be addressed in the discussion. 

Please try to be concise.

_ About S ven metabolites (line 334) is to generic report metabolites and cite a personal comunication.

-in the  sentence line 39-41 the authors reports only one reference (7) related to the increase evidence world wide maybe they need to add other references.

Author Response

Reviewer #2
Sufficient background and relevant references? Can be improved We edited most of the manuscript

References relevant to research? Yes 
Research design appropriate? Can be improved We edited most of the manuscript
Methods described? Can be improved  We edited most of the manuscript
Results clearly presented? Can be improved We edited most of the manuscript

Conclusions supported by results? Yes

Comments:

The manuscript titled "Xanthine dehydrogenase is a modulator of dopaminergic  neurodegeneration in response to bacterial metabolite exposure in C. elegans"  by Thies et al is a  dense  and rich paper in which there are a large amount of  data.

However, the manuscript results difficult to read. In my opinion it is necessary a more systematic organization of  some  paragraphs of the results. The subsection of the results 3.1 contains informations that can be inserted in Materials and Methods sections (the first part related to fig 1). We removed a large section of Results section 3.1 and moved it to the Methods.

Moreover the results are reported in a prolix way with comments and references that may be addressed in the discussion.  Please try to be concise. We changed wording throughout the manuscript to be more concise.

_ About S ven metabolites (line 334) is to generic report metabolites and cite a personal comunication. We are currently working with a natural products pharmacologist on the identification of the active compound(s) within this secondary metabolic extract. While we define the activity of this neurotoxic compound as a “metabolite” it is only a semi-purified mixture. We have removed the Personal Communication reference. Instead, we have added more information in both the Methods and the beginning of the Results section, that define the nature of the metabolite as a mixture.

-in the  sentence line 39-41 the authors reports only one reference (7) related to the increase evidence world wide maybe they need to add other references. Thank you for this suggestion. This has been addressed.

Reviewer 3 Report

Abstract: The abstract was clear and succinctly summarized the rationale and findings. One question that pops into my mind reading the first few sentences is whether exposure to S. ven metabolites reproduces the C. elegans phenotype observed after S. ven exposure. Perhaps this was previously published by the group or will be answered later in the manuscript. But if not, it may be helpful to add this information to Line 14.

Introduction: The introduction and background information is well presented and comprehensive. However, beginning in line 60, they start writing about the previously identified metabolite. I read the original cited paper and found that it was published in 2009, with a follow up in 2014. A little more background information on the characterization of this unknown metabolite would be helpful to the review. It was frustrating to have to read two other papers to get this information. The rest of the introduction is good, and the investigators tell a very interesting scientific story as to how they progressed in their discoveries.

Materials and Methods: Might want to define the CGC acronym in line 101. Presumably that is the Caenorhabditis Genetics Center mentioned 2 lines later.

When describeing the metabolite extraction, it sounds like this is a general extract of lipophilic compounds and not necessarily a purification of a single metabolite, as one would be led to believe by the text and constant referral to "the metabolite". Have the investigators performed NMR or LC-MS on their S. ven extracts to characterize its composition? And if so, is that information reported anywhere? Does further fractionation of the extract, through chromatography for example, result in a specific fraction with neurodegenerative activity? If it is an extract, but not necessarily isolated, purified "metabolite", could the authors modify the text to make this clear to readers without strong organic chemistry backgrounds?

Otherwise, the methods section is very clear, well written and comprehensive. The experimental design is rigorous.

Results: Ok Great, looks like many of the above questions are answered in the first paragraph of the results section. Perhaps consider moving this information to the methods and some may be useful in the introduction as well. 

Otherwise, the results are very thoroughly described and well presented, both in the figures and text.

Maybe not good to use a black fill on figure 6H as the individual data points are note visible. There is also some inconsistency as to whether the authors overlay individual data points on their charts or not. I recommend applying this to all the figures.

Also it appears the authors are trying to maintain a consistent 10-fold scale on all of their gene expression charts, however for genes that are down-regulated it may be good adjust this for easier data visualization, especially xdh-1 in Figure 6D.

Discussion: The discussion is well written and engaging. The conclusion section is also fine. I'm very curious to see where this project goes in the future.

Author Response

Reviewer #3
Sufficient background and relevant references? Yes 

References relevant to research? Yes 
Research design appropriate? Yes 
Methods described? Yes 
Results clearly presented? Yes 

Conclusions supported by results? Yes

Abstract: The abstract was clear and succinctly summarized the rationale and findings. One question that pops into my mind reading the first few sentences is whether exposure to S. ven metabolites reproduces the C. elegans phenotype observed after S. ven exposure. Perhaps this was previously published by the group or will be answered later in the manuscript. But if not, it may be helpful to add this information to Line 14. This is answered later in the main text of the manuscript.

Introduction: The introduction and background information is well presented and comprehensive. However, beginning in line 60, they start writing about the previously identified metabolite. I read the original cited paper and found that it was published in 2009, with a follow up in 2014. A little more background information on the characterization of this unknown metabolite would be helpful to the review. It was frustrating to have to read two other papers to get this information. The rest of the introduction is good, and the investigators tell a very interesting scientific story as to how they progressed in their discoveries. Thank you for the positive feedback on this portion of the text. We have moved information from the beginning of the text into the introduction to provide a more comprehensive background on the characterization and extraction of the S. ven metabolite.

Materials and Methods: Might want to define the CGC acronym in line 101. Presumably that is the Caenorhabditis Genetics Center mentioned 2 lines later. This has been corrected.

When describeing the metabolite extraction, it sounds like this is a general extract of lipophilic compounds and not necessarily a purification of a single metabolite, as one would be led to believe by the text and constant referral to "the metabolite". Have the investigators performed NMR or LC-MS on their S. ven extracts to characterize its composition? And if so, is that information reported anywhere? Does further fractionation of the extract, through chromatography for example, result in a specific fraction with neurodegenerative activity? If it is an extract, but not necessarily isolated, purified "metabolite", could the authors modify the text to make this clear to readers without strong organic chemistry backgrounds? We are currently in the processes of identifying the active component(s) of this metabolite. We are collaborating with a natural products pharmacologist, Dr. Lukasz Ciesla, at The University of Alabama. Through several rounds of fractionation, we have narrowed down the active fraction(s) via LC-MS to a much less complex mixture, that contains less than 10 molecules. However, Streptomyces are notorious for their impressive secondary production, and, as such, we have not yet identified the active molecule(s). Thus, respectfully, it is outside the scope of this manuscript to include these efforts here, as the natural products purification/identification will become a separation manuscript. However, to address your concern, we have clarified our language in the metabolite extraction section of the methods and results sections.

Otherwise, the methods section is very clear, well written and comprehensive. The experimental design is rigorous. Thank you for this feedback.

Results: Ok Great, looks like many of the above questions are answered in the first paragraph of the results section. Perhaps consider moving this information to the methods and some may be useful in the introduction as well. Thank you for this suggestion. We have moved some of this information into the introduction and other parts to the methods.

Otherwise, the results are very thoroughly described and well presented, both in the figures and text. Thank you for this positive feedback.

Maybe not good to use a black fill on figure 6H as the individual data points are note visible. There is also some inconsistency as to whether the authors overlay individual data points on their charts or not. I recommend applying this to all the figures. Thank you for this suggestion. The black fill has been changed in the appropriate figures. Additionally, we have addressed the inconsistency with the data point overlays as required. qPCR graphs do not contain data point overlays because these data are pooled, which is different from our neurodegeneration, which were entered into our statistical analysis program (Prism), in triplicate.

Also it appears the authors are trying to maintain a consistent 10-fold scale on all of their gene expression charts, however for genes that are down-regulated it may be good adjust this for easier data visualization, especially xdh-1 in Figure 6D. Thank you for this suggestion. This has been corrected in the appropriate figures.

Discussion: The discussion is well written and engaging. The conclusion section is also fine. I'm very curious to see where this project goes in the future. Thank you for the positive feedback on this manuscript.

Round 2

Reviewer 2 Report

The manuscript is improved.